# PROSPERITY BEFORE COLLAPSE: HOW FAR CAN OFF-POLICY RL REACH WITH STALE DATA ON LLMS?

**Haizhong Zheng**[1]    **Jiawei Zhao**[2]    **Beidi Chen**[1]
[1]Carnegie Mellon University    [2]Meta AI
{haizhonz, beidic}@cmu.edu, jwzhao@meta.com

## ABSTRACT

Reinforcement learning has been central to recent advances in large language model reasoning, but most algorithms rely on on-policy training that demands fresh rollouts at every update, limiting efficiency and scalability. Asynchronous RL systems alleviate this by decoupling rollout generation from training, yet their effectiveness hinges on tolerating large staleness in rollout data, a setting where existing methods either degrade in performance or collapse. We revisit this challenge and uncover a *prosperity-before-collapse* phenomenon: stale data can be as informative as on-policy data if exploited properly. Building on this insight, we introduce **M2PO** (Second-Moment Trust Policy Optimization), which constrains the second moment of importance weights to suppress only extreme outliers while preserving informative updates. Notably, M2PO sharply reduces the fraction of clipped tokens under high staleness (from 1.22% to 0.06% over training), precisely masking high-variance tokens while maintaining stable optimization. Extensive evaluation across six model scales (from 1.7B to 32B) and eight math reasoning benchmarks and one coding benchmarks shows that M2PO delivers stable off-policy training even with data stale by *at least 256 model updates* and matches on-policy performance. Our code is available at https://github.com/Infini-AI-Lab/M2PO/.

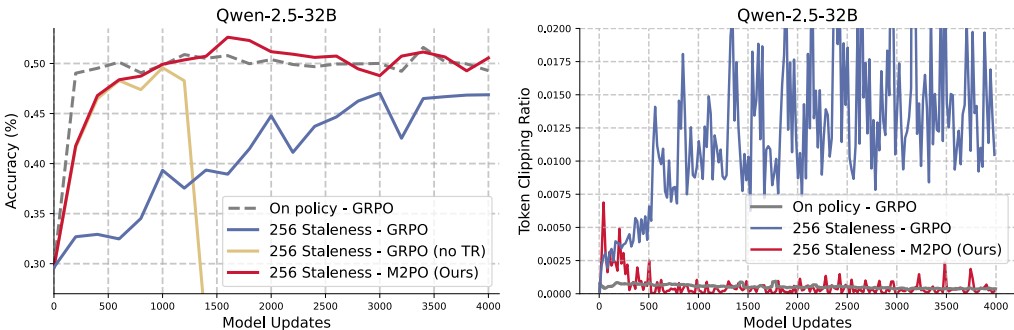

Figure 1: Comparison of on-policy GRPO and off-policy training under a staleness of 256 model updates on Qwen-2.5-32B. **Left:** Standard GRPO suffers from degradation with stale rollouts, while removing the trust region (GRPO no TR) reveals a clear *prosperity-before-collapse* phenomenon. In contrast, M2PO achieves stable training and matches on-policy performance even under high staleness. **Right:** Token clipping ratio comparison shows that M2PO dramatically reduces clipping events compared to GRPO with the same staleness, while avoiding training collapse.

## 1 INTRODUCTION

Reinforcement learning (RL) has been central to recent advances in large language model (LLM) reasoning, driving breakthroughs in systems like OpenAI's o1 (OpenAI et al., 2024) and DeepSeek's R1 (DeepSeek-AI et al., 2025; Team et al., 2025). Most existing RL algorithms (Schulman et al., 2017; Zheng et al., 2025b; Yu et al., 2025) for LLMs adopt an on-policy design, as it provides stable training and reliable performance, but the strict requirement for fresh (or limited-staleness) rollouts

at every update. This constraint significantly limits scalability and becomes increasingly impractical as we move toward harder, more complex reasoning or agentic tasks, where rollouts involve multi-step tool use, expensive program execution, and long reward computation. In such settings, rollout latency can easily range from minutes to hours, making substantial staleness not only common but often unavoidable. For example, SWE-bench Jimenez et al. (2023) with OpenHands Wang et al. (2024), rollout latency can become extremely large. In a single run, even with a small batch size, the end-to-end inference time (including tool call and code execution) can exceed 100 minutes, with more than 80 iterations in the environment, which makes on-policy training extremely inefficient. To overcome this bottleneck, a growing line of RL systems (Fu et al., 2025; Zhu et al., 2025; Noukhovitch et al., 2024; Zhong et al., 2025; He et al., 2025) have explored asynchronous designs that decouple rollout from training. Such approaches improve resource utilization and enable training to scale more efficiently across large and heterogeneous clusters, but their effectiveness fundamentally relies on the ability of RL algorithms to tolerate rollout staleness without sacrificing stability or performance.

However, under large rollout staleness, existing RL algorithms struggle to strike the right balance. Some methods (Schulman et al., 2017; Shao et al., 2024; Zheng et al., 2025a) can maintain stability, but they often suffer from noticeable performance degradation. Conversely, approaches designed to maximize performance (Fu et al., 2025; Chen et al., 2025; Su et al., 2025) tend to compromise stability, frequently leading to training collapse. On-policy methods provide both stability and strong performance, but their reliance on fresh or only slightly stale rollouts at every update imposes rigid constraints that hinder scalability. Consequently, an ideal off-policy RL algorithm for LLMs should enable effective reuse of trajectories collected under outdated policies to preserve strong performance under significant staleness, and ensure stable training that converges competitively with on-policy methods. Meeting these requirements is key to realizing off-policy RL as a truly scalable solution for aligning and fine-tuning large language models.

In this paper, we aim to investigate the underlying reasons for the limitations of off-policy RL in LLMs and to design an effective algorithm that fully leverages stale data to unlock its potential. We begin by revealing an intriguing *Prosperity before Collapse* phenomenon (Yellow curve in Figure 1 (left)): although RL training without a trust region eventually collapses on stale data, it initially achieves substantially higher performance than vanilla GRPO with $\epsilon$-clipping. In some cases, it even matches the performance of the on-policy baseline. From this, we draw an important observation: *stale data can be as informative as data collected on-policy in RL for LLMs*, but the key challenge lies in how existing algorithms exploit it. In particular, vanilla GRPO performs poorly under staleness because stale-data training exhibits a substantially higher clipping rate, with many of the clipped updates occurring on informative high-entropy tokens (see Figure 4). This disproportionate clipping on crucial tokens hinders the full utilization of stale training data.

This pivotal token masking observation reveals that these high-entropy tokens play a dual role: they provide the most informative training signal but also introduce the greatest instability under staleness. Therefore, the key challenge is to retain as much learning signal from these tokens as possible without risking training collapse. Motivated by this, we propose **M2PO** (Second-Moment Trust Policy Optimization), a novel off-policy RL algorithm that constrains the second moment of importance weights. Unlike standard $\epsilon$-clipping, which disproportionately suppresses high-entropy tokens and discards valuable learning signals, M2PO leverages the second-moment metric $M_2$. This metric is both variance-sensitive, capturing instability introduced by high-entropy tokens, and statistically stable, avoiding the cancellation issues inherent to KL-based measures. By regularizing training at the batch level through $M_2$, M2PO masks only extreme outliers while preserving the majority of informative updates. As a result, M2PO enables stable off-policy reinforcement learning with stale data, matching on-policy performance even under large staleness.

As illustrated in Figure 1 (left), even when trained exclusively on data stale by at least 256 model updates, M2PO achieves accuracy comparable to the on-policy baseline (red curve), demonstrating its ability to fully exploit stale data without sacrificing stability. M2PO achieves this through a more accurate and adaptive clipping strategy that clips substantially fewer tokens while maintaining training stability. As shown in Figure 1 (right), M2PO dramatically reduces the fraction of clipped tokens under high staleness (from 1.22% to 0.06% over the entire training process, see Figure 6d), thereby preserving more useful training information in stale data. To further validate M2PO effectiveness, we conduct an extensive evaluation of M2PO across six model scales (ranging from 1.7B to 32B) and eight math reasoning benchmarks in Section 6. The results show that M2PO consistently deliv-

ers strong performance across all training settings. M2PO also shows insensitivity to the choice of threshold, with a single value across all experiments, demonstrating its practicality and robustness.

## 2 RELATED WORK

**RLVR.** Recent advances (DeepSeek-AI et al., 2025; Yu et al., 2025; Team et al., 2025; Gao et al., 2024) in LLM reasoning show that Reinforcement Learning with Verifiable Reward (RLVR), which relies on verifiable reward signals instead of model-generated scores, can effectively improve model reasoning ability. These gains are achieved using various policy optimization methods such as PPO (Ouyang et al., 2022) and GRPO (Shao et al., 2024). Encouraged by the success of RLVR, a growing body of work (Kazemnejad et al., 2024; Yuan et al., 2025b;a; Yu et al., 2025; Liu et al., 2025b; Luo et al., 2025a; Zhang et al., 2025; Hu, 2025; Xiong et al., 2025) has emerged to further improve reinforcement learning methods for LLM reasoning. For instance, methods such as VinePPO (Kazemnejad et al., 2024), VC-PPO (Yuan et al., 2025b), and VAPO (Yuan et al., 2025a) aim to enhance LLM reasoning by optimizing the value function.

**Trust Region in RLVR.** While RLVR has been widely adopted for fine-tuning LLMs, a key challenge lies in how to effectively constrain the trust region, not only to stabilize training but also to achieve better learning efficiency and overall performance. To address this, a growing line of work has proposed various strategies to control the policy update, ranging from ratio clipping (Yu et al., 2025), approximate trust region (Fu et al., 2025), sequence-level clipping (Zheng et al., 2025a), asymmetric trust region (Roux et al., 2025; Arnal et al., 2025), and gradient-preserving clipping (Su et al., 2025; Chen et al., 2025). For instance, AREAL (Fu et al., 2025) uses a more recent approximate policy to decide the trust region rather than the behavior model. GSPO (Zheng et al., 2025a) moves from token-level to sequence-level clipping by defining importance ratios on sequence likelihood. While these methods improve RLVR under moderate settings, most of them focus on relatively limited intra-iteration staleness (e.g., 8 or 16) and have not been thoroughly studied under larger off-policy gaps, like extreme staleness. In this work, our goal is to better understand the role of staleness in RLVR and to seek more effective ways of constraining the trust region in RLVR.

## 3 BACKGROUND

### 3.1 GROUP RELATIVE POLICY OPTIMIZATION (GRPO)

Group Relative Policy Optimization (GRPO) (Shao et al., 2024) is a variant of Proximal Policy Optimization (PPO) (Ouyang et al., 2022) tailored for language model fine-tuning. Instead of computing advantages using a value function, GRPO normalizes reward scores within groups of responses sampled for the same prompt, which largely improves the training efficiency, and aims to maximize the following objective:

$$\mathcal{J}_{GRPO}(\theta) = \mathbb{E}[q \sim P(Q), \{o_i\}_{i=1}^G \sim \pi_{\theta_{old}}(O|q)]$$

$$\frac{1}{G} \sum_{i=1}^G \left( \min \left( \frac{\pi_\theta(o_i|q)}{\pi_{\theta_{behav}}(o_i|q)} A_i, \text{clip}\left( \frac{\pi_\theta(o_i|q)}{\pi_{\theta_{behav}}(o_i|q)}, 1-\epsilon, 1+\epsilon \right) A_i \right) \right), \tag{1}$$

where $A_i$ is the advantage, computed using a group of rewards $\{r_1, r_2, \ldots, r_G\}$ corresponding to the outputs within each group:

$$A_{i,t} = \frac{r_i - \text{mean}(\{R_i\}_{i=1}^G)}{\text{std}(\{R_i\}_{i=1}^G)}. \tag{2}$$

Similar to PPO, GRPO employs a clipping mechanism to stabilize updates. The ratio $r_i = \frac{\pi_\theta(o_i|q)}{\pi_{\theta_{old}}(o_i|q)}$ is clipped to $[1-\epsilon, 1+\epsilon]$, so that when $A_i > 0$ the policy cannot increase probability mass excessively, and when $A_i < 0$ it cannot over-penalize. This prevents large, unstable updates while still allowing normalized group advantages to guide learning.

## 3.2 PERFORMANCE DEGRADATION FROM TRAINING WITH STALE DATA

To investigate the impact of stale data on reinforcement learning for large language models, we introduce Stale-$k$ RL training, where the model is trained using data generated $k$ model updates earlier in each training iteration. More specifically, in our training setup, each training step consists of four model updates, a configuration commonly used in recent work (Zheng et al., 2025b; Wang et al., 2025b; Yu et al., 2025; Chen et al., 2025; Zheng et al., 2025a). Thus, even stale-0 ($s$=0) training has a staleness between 0 and 3. stale-256 ($s$=256) training has a staleness between 256 and 259). During the first $k$ model updates, since no stale model is yet available, the model is trained on data generated by the original base model, with different training data used in each iteration. In this setup, all training data after the initial phase comes from stale models, allowing us to study how stale data affects the dynamics and effectiveness of RL training.

**Stale-$k$ RL training.** In our "stale-k" setup, the training batch at step $t$ is collected $k$ updates earlier, but we implement this by storing the generated data rather than storing old checkpoints. Concretely, at update step $t - k$ we use the current policy to generate rollouts and place them into a buffer, and after $k$ further updates, these trajectories are consumed once for training at step $t$. Each sampled trajectory is used exactly once and is never reused across multiple updates. More training details can be found at Appendix B.

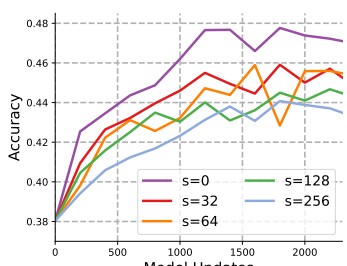

As shown in Figure 2, we train Qwen2.5-Math-7B (Yang et al., 2024) with GRPO under varying staleness levels and report test accuracy. The results reveal a clear trend: as staleness increases, model performance degrades and convergence slows. In particular, low-staleness training achieves higher accuracy, whereas high-staleness training converges more slowly to lower performance.

Figure 2: RL with stale data on Qwen2.5-Math-7B, reporting avg. accuracy on all benchmarks.

## 4 PROSPERITY BEFORE COLLAPSE: STALE DATA CONTAIN ENOUGH TRAINING INFORMATION IN RL ON LLMS

In this section, we investigate why RL on LLM deteriorates when trained on stale data generated by earlier policies.

First, we reveal an intriguing *prosperity-before-collapse* phenomenon: although off-policy RL training without a trust region eventually collapses on stale data, it achieves substantially higher performance than GRPO with $\epsilon$-clipping before collapse, even matching on-policy results. Next, we study the causes of GRPO's inferior performance when trained with stale data.

**Prosperity before collapse: training without a trust region.** To disentangle whether the performance drop stems from stale data generated by highly shifted old policies or from biases introduced by the training algorithm, we remove the trust region entirely to remove bias from the training algo-

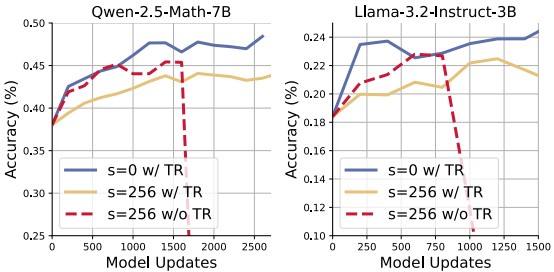

Figure 3: Prosperity before Collapse. Training without a trust region (TR) ($\epsilon = \infty$) under stale data ($s = 256$) initially achieves higher performance than clipped training, sometimes even matching the on-policy baseline ($s = 0$). However, it eventually collapses due to uncontrolled variance.

rithm, we remove the trust region entirely to remove bias from the training algorithm. Surprisingly, we observe a distinct *prosperity-before-collapse* phenomenon. As shown in Figure 1 and Figure 3, although training without a trust region eventually collapses, it achieves substantially better performance prior to collapse. In fact, under stale data ($s$=256), the no-clipping setting initially outperforms clipped training, sometimes even matching on-policy baselines.

**Pivotal token masking by $\epsilon$-clipping when training with stale data.** As also discussed in recent work (Su et al., 2025; Chen et al., 2025), $\epsilon$-clipping may inadvertently mask important tokens,

preventing them from contributing useful training signals. We extend this observation to the asynchronous setting and show that the problem becomes substantially more severe when training with stale data, since larger staleness induces a greater mismatch between the behavior and target policies. As illustrated in Figure 4a, the clipping ratio increases sharply under large staleness ($s = 256$), while remaining negligible in the on-policy baseline.

To better understand this phenomenon, we conduct a quantitative analysis on 90 million training tokens collected during Qwen2.5-Math-7B training with staleness 256. Specifically, we gather all training tokens generated between 800 and 1200 model updates, ensuring the model is already in a stable training phase but before convergence. Figure 4b shows a clear trend: as $|r - 1|$ increases, the average token entropy also rises.

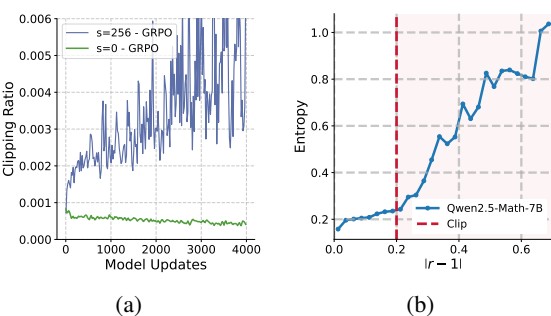

(a)                                      (b)

Figure 4: **(a)** Clipping ratio dynamics during RL training on the Qwen-2.5-Math-7B model. **(b)** Relationship between average token entropy and the distance between the importance sampling ratio $r$ and 1.

This indicates that $\epsilon$-clipping disproportionately prunes high-entropy tokens, which are typically the most informative for model improvement (Wang et al., 2025a; Gao et al., 2025; Cui et al., 2025). Consequently, clipping under stale data leads to degraded performance.

This observation reveals a dilemma: while high-entropy tokens are crucial for learning progress, they also introduce instability in the off-policy setting, which motivates our key research question:

> *Can a more accurate and adaptive trust region strategy preserve the benefits of stale data while ensuring stable training?*

## 5   SECOND-MOMENT TRUST POLICY OPTIMIZATION

In this section, we propose Second-Moment Trust Policy Optimization (M2PO), a novel policy optimization algorithm providing a more effective trust region for off-policy training with stale data. As discussed in Section 4, as high-entropy tokens are a double-edged sword, the key challenge in designing an effective trust region algorithm is how to best harness the rich information in high-entropy tokens without letting them destabilize training.

### 5.1   MEASURING DISTRIBUTION GAP WITH THE SECOND MOMENT

The main source of instability in off-policy RL lies in the distributional mismatch between the behavior policy that generates training data and the current policy being optimized (Schulman et al., 2015; 2017). As the divergence between these two distributions grows, importance sampling corrections produce high-variance gradient estimates, leading to noisy and unreliable updates. Our motivation is therefore to constrain the distributional gap between $\pi_{\text{behav}}$ and $\pi_\theta$ at the batch level, directly coupling the constraint with model updates while preventing over-constraining of token-level variations.

A natural choice to measure distribution is the batch-level KL divergence, a metric widely adopted to monitor stability in RL:

$$\hat{KL} = \frac{1}{N} \sum_{i=1}^{N} \hat{KL}_i = -\frac{1}{N} \sum_{i=1}^{N} \log r_i = -\frac{1}{N} \sum_{i=1}^{N} \log \frac{\pi_\theta(a_i \mid s_i)}{\pi_{\text{behav}}(a_i \mid s_i)}, \tag{3}$$

where $N$ is the number of tokens in a batch.

However, batch-level KL suffers from two key limitations. First, because it is computed from single-sample estimates, individual $\hat{KL}_i$ can be positive or negative, leading to *cancellation effects* where

large deviations offset each other and produce deceptively small KL values. Second, tokens with large ratios ($r_i > 1$) are not properly constrained, as their negative $\hat{KL}_i$ actually decreases the estimated KL, even though such tokens can contribute to training instability (Schulman et al., 2017).

To overcome these limitations, we propose to use the second moment of the log-ratio to measure the distribution gap between behavior and current policy. Formally, we define

$$\hat{M}_2 = \frac{1}{N}\sum_{i=1}^{N}\hat{M}_{2,i} = \frac{1}{N}\sum_{i=1}^{N}(\log r_i)^2 = \frac{1}{N}\sum_{i=1}^{N}[\log\frac{\pi_\theta(a_i \mid s_i)}{\pi_{\text{behav}}(a_i \mid s_i)}]^2, \tag{4}$$

This choice is motivated by two key advantages of $\hat{M}_2$ over the batch $\hat{KL}$. First, each per-token estimate $\hat{M}_{2,i} = (\log r_i)^2$ is always non-negative, so the constraint can be reliably applied even when $r > 1$. Second, while the batch KL only measures the mean shift between policies, $M_2$ also reflects the variance of importance weights. This makes $\hat{M}_2$ more sensitive to outliers and noisy tokens with extreme ratios $r_i$[1].

Furthermore, Theorem 1 shows that although $M_2$ does not directly constrain $r - 1$ like $\epsilon$ clipping, it nevertheless provides an upper bound on the Pearson chi-square divergence $\mathbb{E}[(r-1)^2]$ between the new and behavior policies. The proof is provided in Appendix D.

**Theorem 1** (Bounding $\chi^2$ by $M_2$). *Let $r = \frac{\pi_{\text{new}}}{\pi_{\text{behav}}}$ be the importance ratio and assume $1/R \leq r \leq R$. Define the log-ratio second moment*

$$M_2 = \mathbb{E}_{a\sim\pi_{\text{behav}}}\big[(\log r(a))^2\big].$$

*Let the Pearson chi-square divergence between $\pi_{\text{new}}$ and $\pi_{\text{behav}}$ be*

$$\chi^2(\pi_{\text{new}}\,\|\,\pi_{\text{behav}}) = \mathbb{E}_{a\sim\pi_{\text{behav}}}\left[\left(\frac{\pi_{\text{new}}(a)}{\pi_{\text{behav}}(a)} - 1\right)^2\right] = \mathbb{E}_{\pi_{\text{behav}}}\big[(r-1)^2\big].$$

*Then*

$$\chi^2(\pi_{\text{new}}\,\|\,\pi_{\text{behav}}) \;\leq\; R^2\,M_2.$$

## 5.2 SECOND-MOMENT TRUST POLICY OPTIMIZATION

As illustrated in Algorithm 1, to maintain training stability, M2PO applies a masking strategy that selectively excludes tokens until the batch-level $\hat{M}_2$ of the remaining tokens falls below a predefined threshold $\tau_{M_2}$. Importantly, we observe that $\tau_{M_2}$ is not a sensitive hyperparameter (see Figure 7). Across all our experiments, we consistently set $\tau_{M_2} = 0.04$, and this single setting proved effective for stabilizing training in all training scenarios.

**Only constrain trust-region tokens.** Although the PPO loss clips the ratio on both the upper and lower sides, due to the use of the `min` operator, not all tokens are actually clipped. In practice, clipping only occurs for tokens where $A > 0$ and $r > 1$, or $A < 0$ and $r < 1$. Following the PPO setting, we therefore apply the $M_2$ constraint exclusively to tokens that satisfy these conditions. Finally, with the result mask $M$, we update the policy by maximizing the following objective[2]:

---

**Algorithm 1:** M2PO Masking

**Input:** $\{\hat{M}_{2,i}\}_{i=1}^{N}$ for all training tokens; threshold $\tau_{M_2}$
**Output:** mask $M$

1   $M \leftarrow$ `True` for all tokens;
2   $\mathcal{T} \leftarrow$ all trust-region tokens;
3   /* Sort:  O(NlogN)     */
4   Sort all tokens w.r.t. $\hat{M}_{2,i}$.
5   **while** $\text{mean}_{i\in\mathcal{T}}\,\hat{M}_{2,i} > \tau_{M_2}$ **do**
6      /* Select:  O(1)     */
7      $j \leftarrow \arg\max_{i\in\mathcal{T}}\,\hat{M}_{2,i}$;
8      $M_j \leftarrow$ `False`;
      $\mathcal{T} \leftarrow \mathcal{T}\setminus\{j\}$;
9   **end while**
10 **return** $M$

---

[1] A potential alternative is to use $\sum_{i=1}^{N}|\hat{KL}_i|/N$. While this absolute KL estimate can also work empirically, it is less sensitive to variance compared to $M_2$ Moreover, M2 provides an upper bound for this absolute KL estimate, as $E[|r|] \leq \sqrt{E[r^2]}$. Therefore, we adopt $M_2$ in our method.

[2] In our loss, we average over all tokens rather than only the unmasked ones. This choice is intended to better mimic the behavior of PPO-style clipping. However, since the masking ratio is typically very small (see Section 6.3), the difference between the two averaging strategies is negligible in practice.

$$\mathcal{J}_{\text{M2PO}}(\theta) = \frac{1}{\sum_{i=1}^{G}|o_i|} \sum_{i=1}^{G} \sum_{t=1}^{|o_i|} \boldsymbol{M}_{i,t} \frac{\pi_\theta(o_i|q)}{\pi_{\theta_{behav}}(o_i|q)} A_{i,t}, \qquad \boldsymbol{M}_{i,t} \in \{0,1\}, \tag{5}$$

where $A_i$ denotes the advantage, computed using the grouped advantage in Equation 2.

## 6 EXPERIMENTS

In this section, we present an extensive evaluation across six models (from 1.7B to 32B) on eight benchmarks. The results demonstrate that, even when trained with extremely stale data, M2PO achieves performance comparable to on-policy GRPO and significantly outperforms other baselines:

- In Section 6.2, we show that M2PO achieves **accuracy on par with on-policy baselines** under large staleness ($s = 256$), and outperforms baselines by up to 11.2% in average accuracy.
- In Section 6.3, we provide a detailed analysis of how M2PO boosts off-policy RL performance while preserving training stability. We also show that its sole threshold hyperparameter $\tau_{M_2}$ is insensitive to variation, ensuring ease of use in practice.

### 6.1 EXPERIMENTAL SETTINGS

**Models & Datasets.** To verify the effectiveness of our method, we extensively evaluate M2PO on six models: Qwen2.5-Math-7B (Yang et al., 2024), Llama-3.2-3B-Instruct (Dubey et al., 2024), Qwen3-Base-1.7B/4B/8B (Yang et al., 2025), and Qwen2.5-32B (Yang et al., 2024). For Qwen2.5-Math-7B, we use a context length of 4k, which is the maximum for this series, while for all other models the context length is set to 16k. For training, we adopt the DeepScaleR (Luo et al., 2025b) math dataset.

**Training & Evaluation.** Our method is implemented based on verl (Sheng et al., 2024) pipeline and uses vLLM (Kwon et al., 2023) for rollout. We use a mix of H100 and H200 servers for training, depending on resource availability. For benchmark datasets, we use eight widely used complex mathematical reasoning benchmarks to evaluate the performance of trained models: Math500 (Hendrycks et al., 2021; Lightman et al., 2023), AIME24/25 (Art of Problem Solving, 2024a), AMC23/24 (Art of Problem Solving, 2024b), Minerva Math (Lewkowycz et al., 2022), Gaokao (Zhang et al., 2023), Olympiad Bench (He et al., 2024). Similar to (Wang et al., 2025b; Zheng et al., 2025b), we evaluate models on those benchmarks every 50 steps and report the performance of the checkpoint that obtains the best average performance on eight benchmarks. For GRPO, we adopt the commonly used clipping parameter $\epsilon = 0.2$, while for the other baselines, we follow the recommended values reported in their respective papers. We include more detailed experimental settings in Appendix B.

### 6.2 PERFORMANCE COMPARISON ON TRAINING WITH STALENESS

**Prosperity without collapse: Stable off-policy training without performance degradation using M2PO.** To verify the effectiveness of M2PO, Table 1 presents a comprehensive comparison of math reasoning performance across eight benchmarks using models from four different families and scales, ranging from 1.7B to 32B parameters. We evaluate multiple reinforcement learning methods under both on-policy and off-policy settings, including GRPO, GSPO, and our proposed M2PO. The results show that while both GRPO and GSPO often suffer significant performance drops under large staleness, M2PO consistently achieves comparable accuracy to the on-policy baseline in all training settings. Surprisingly, we notice that, in some model settings, M2PO with $s = 256$ even achieves a better performance than M2PO with $s = 0$. For instance, on the Qwen3-Base-1.7B model, we observe that M2PO with $s = 256$ (36.6%) outperforms GRPO with $s = 0$ (33.0%). A potential explanation is that small effective staleness (e.g., $s = 0$ corresponding to delays between 0 and 3) can still adversely affect training stability. Our further analysis in Figure 6 supports this view, showing that M2PO with $s = 256$ exhibits an even lower clipping ratio than GRPO with $s = 0$. Overall, these results

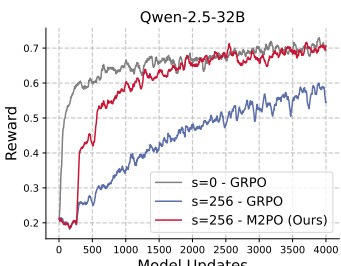

Figure 5: Training reward on Qwen-2.5-32B.

Table 1: Performance (%) comparison across eight math reasoning benchmarks using models from 1.7B to 32B parameters. We report results for GRPO, GSPO, and M2PO under both on-policy ($s = 0$) and off-policy ($s = 256$) settings. Underlined numbers denote the best average accuracy, while **bold** numbers highlight the best average accuracy under stale rollouts ($s = 256$). M2PO consistently improves stability under staleness and achieves higher average accuracy than GRPO.

| Method | S | AIME24/25 | AMC23/24 | Math500 | Gaokao | Miner. | Olymp. | Avg. |
|---|---|---|---|---|---|---|---|---|
| *Llama-3.2-3B-Instruct* | | | | | | | | |
| GRPO | 0 | 11.0 / 2.3 | 31.3 / 17.2 | 53.6 | 42.99 | 23.1 | 20.3 | 25.2 |
| GRPO | 256 | 9.6 / 0.4 | 25.0 / 13.9 | 52.4 | 42.08 | 17.6 | 18.8 | 22.5 |
| GSPO | 256 | 9.0 / 0.2 | 30.0 / 14.4 | 50.6 | 40.65 | 18.8 | 17.3 | 22.6 |
| M2PO (Ours) | 256 | 10.4 / 4.4 | 33.8 / 17.8 | 52.0 | 44.48 | 21.2 | 18.1 | **25.3** |
| *Qwen2.5-Math-7B* | | | | | | | | |
| GRPO | 0 | 39.6 / 17.5 | 63.8 / 46.7 | 82.3 | 64.1 | 36.7 | 43.6 | 49.3 |
| GRPO | 256 | 29.4 / 12.9 | 64.4 / 39.4 | 80.5 | 63.2 | 33.1 | 43.1 | 45.7 |
| GSPO | 256 | 27.3 / 13.1 | 63.8 / 36.7 | 79.0 | 62.2 | 33.5 | 41.9 | 44.7 |
| M2PO (Ours) | 256 | 33.3 / 17.5 | 63.8 / 40.6 | 84.0 | 66.4 | 38.1 | 47.1 | **48.8** |
| *Qwen3-Base-1.7B* | | | | | | | | |
| GRPO | 0 | 7.5 / 7.5 | 40.6 / 26.1 | 67.2 | 55.9 | 28.9 | 30.5 | 33.0 |
| GRPO | 256 | 8.5 / 4.8 | 34.4 / 25.0 | 64.3 | 52.7 | 26.0 | 27.6 | 30.4 |
| GSPO | 256 | 6.9 / 4.0 | 39.4 / 18.9 | 65.0 | 53.1 | 26.5 | 27.5 | 30.1 |
| M2PO (Ours) | 256 | 14.0 / 6.5 | 48.1 / 27.8 | 71.8 | 59.5 | 29.4 | 35.6 | **36.6** |
| *Qwen3-Base-4B* | | | | | | | | |
| GRPO | 0 | 22.9 / 20.2 | 63.8 / 53.9 | 84.6 | 69.8 | 40.2 | 50.5 | 50.7 |
| GRPO | 256 | 14.0 / 9.6 | 51.9 / 32.8 | 76.8 | 61.7 | 34.4 | 39.8 | 40.1 |
| GSPO | 256 | 17.9 / 15.4 | 55.6 / 38.3 | 76.8 | 62.3 | 35.1 | 44.3 | 43.2 |
| M2PO (Ours) | 256 | 26.7 / 21.0 | 64.4 / 49.4 | 85.8 | 70.3 | 40.5 | 52.3 | **51.3** |
| *Qwen3-Base-8B* | | | | | | | | |
| GRPO | 0 | 26.7 / 19.4 | 76.9 / 52.8 | 87.7 | 71.6 | 41.2 | 52.8 | 53.6 |
| GRPO | 256 | 21.0 / 13.1 | 63.8 / 40.0 | 81.8 | 67.8 | 38.5 | 47.4 | 46.7 |
| M2PO (Ours) | 256 | 30.2 / 23.1 | 71.3 / 56.7 | 87.2 | 75.1 | 42.6 | 54.8 | **55.1** |
| *Qwen2.5-32B* | | | | | | | | |
| GRPO | 0 | 24.4 / 18.3 | 71.9 / 46.7 | 85.4 | 71.9 | 41.4 | 52.9 | 51.6 |
| GRPO | 256 | 20.4 / 9.6 | 68.1 / 41.1 | 83.0 | 67.3 | 40.9 | 45.9 | 47.0 |
| M2PO (Ours) | 256 | 24.8 / 19.4 | 76.3 / 50.0 | 85.7 | 71.7 | 41.5 | 51.7 | **52.6** |

show that M2PO remains robust and effective, sustaining stable, high performance even under extreme off-policy conditions.

In addition to the final accuracy comparison in Table 1, we also analyze the training dynamics of accuracy and reward of Qwen-2.5-32B models. As shown in Figure 1, M2PO with $s = 256$ initially falls behind the on-policy baseline but quickly catches up, eventually matching its performance, while converging much faster and achieving higher accuracy than GRPO under the same staleness. This highlights that M2PO not only maintains comparable final accuracy but also accelerates convergence when training with stale data. Figure 5 shows a similar trend in the reward curves. M2PO with $s = 256$ also starts off behind the on-policy baseline due to the initial plateau caused by using data generated from the base model, but it quickly catches up and aligns closely with the $s = 0$ trajectory. In contrast, GRPO with $s = 256$ consistently underperforms across the entire training trajectory.

**Performance of other baselines under staleness.** A number of prior works have proposed alternative trust region strategies beyond $\epsilon$-clipping, including GSPO (Zheng et al., 2025a), AREAL (Fu et al., 2025), TOPR (Roux et al., 2025), GPPO (Su et al., 2025), and CISPO (Chen et al., 2025). *Despite not being designed to handle the extreme staleness studied in this paper*, we also evaluate these methods under our setting with $s = 256$ for completeness and comparison. Among these methods, GSPO is the only one that preserves training stability, though it still exhibits a noticeable

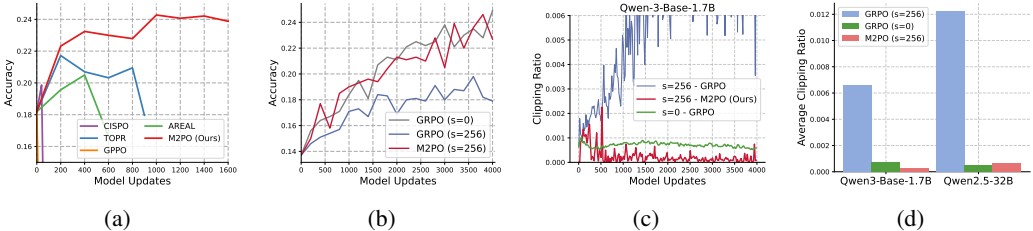

Figure 6: **(a)** Methods comparison under staleness ($s = 256$) on Llama3.2-Instruct-3B. **(b)** Performance comparison between M2PO and GRPO on coding tasks. **(c)** Clipping ratio dynamics during RL on the Qwen-3-Base-1.7B model. **(d)** Comparison of the average clipping ratio across models and methods.

performance drop under high staleness (see Table 1). For the other methods, as shown in Figure 6a, most encounter substantial difficulties in maintaining training stability and tend to break down early in training. These observations suggest that while existing approaches can be effective under moderately stale settings, they face significant challenges when extended to larger staleness, highlighting the need for a more robust and effective solution.

**Evaluation results on coding tasks.** We further evaluate M2PO on coding tasks beyond the math domain. Specifically, we train **DeepSeek-R1-Distill-Qwen-1.5B** DeepSeek-AI et al. (2025) on the `code_contests` dataset (Li et al., 2022) and evaluate on `LiveCodeBench` (Jain et al., 2024). We follow a training setup similar to that used for math tasks, except that each training iteration samples a batch of 128 prompts for rollout and 32 prompts for each model updates. We report **avg@4** accuracy as the evaluation metric.

As shown in Figure 6b, M2PO with staleness $s = 256$ outperforms GRPO at the same staleness level and achieves performance comparable to GRPO with $s = 0$, consistent with our observations on math reasoning tasks. This further demonstrates the generality of M2PO on other tasks beyond math.

## 6.3 ANALYSIS AND ABLATION STUDY

**Stable training with reduced clipping.** Figure 6(a)(b) illustrates the clipping dynamics of GRPO and our proposed M2PO under different staleness settings. In Figure 6c, we report results on Qwen-3-Base-1.7B. Under large staleness ($s = 256$), GRPO exhibits frequent clipping events, with the ratio increasing sharply and remaining high during most of the training. In contrast, M2PO under the same staleness maintains an exceptionally low clipping ratio, comparable to or even lower than the on-policy GRPO baseline ($s = 0$). Notably, M2PO with $s = 256$ exhibits less clipping than GRPO with $s = 0$, which explains why M2PO with $s = 256$ achieves higher accuracy than GRPO with $s = 0$ in Table 1. Figure 1 shows the same comparison on Qwen-2.5-32B. A similar trend holds: GRPO with $s = 256$ suffers from substantial clipping, whereas M2PO effectively suppresses unnecessary clipping, remaining close to the on-policy baseline.

Figure 6d summarizes the average clipping ratio across the entire training process. On Qwen-3-Base-1.7B, GRPO with $s = 256$ reaches an average clipping ratio of 0.66%, compared to 0.07% for GRPO with $s = 0$ and only 0.02% for M2PO with $s = 256$. On Qwen-2.5-32B, GRPO with $s = 256$ averages 1.22%, while GRPO with $s = 0$ records 0.05% and M2PO with $s = 256$ maintains a similarly low ratio of 0.06%. These results show that M2PO reduces clipping by over an order of magnitude compared to GRPO, thereby enabling stable and efficient training by clipping only when necessary.

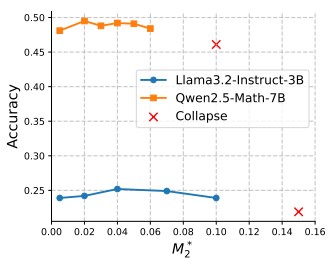

**Robustness to the choice of $\tau_{M_2}$.** Figure 7 shows that performance is not highly sensitive to the choice of $\tau_{M_2}$. Accuracy remains stable across a broad range, only dropping when $\tau_{M_2}$ is set extremely small (overly restrictive constraint) or very large (train-

Figure 7: Ablation study of the $\tau_{M_2}$ threshold on Llama-3.2-3B-Instruct and Qwen2.5-Math-7B.

ing collapse). This explains why a single setting of $\tau_{M_2} = 0.04$ works robustly across all training settings in our paper.

**Training dynamics on KL and $M_2$.** Figure 8 shows the impact of M2PO masking on training stability. Figure 8a shows that the average $M_2$ without masking exhibits frequent spikes throughout training, indicating instability in the second-moment estimates (blue curve). Applying M2PO masking effectively suppresses these fluctuations and maintains consistently low $M_2$ values, leading to more stable updates (red curve). Figure 8b compares the KL divergence across different methods. Although M2PO with $s = 256$ involves substantially less clipping than GRPO (shown in Figure 6), it maintains a more stable divergence than GRPO with $s = 256$. These results indicate that M2PO performs clipping in a more precise and adaptive manner, ensuring training stability with substantially less reliance on clipping. These results demonstrate that M2PO enables more precise and adaptive clipping, achieving training stability while relying on significantly fewer clipping operations, and thereby attaining better performance without risking training collapse.

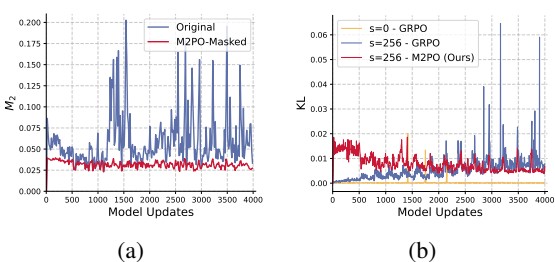

(a)  (b)

Figure 8: (a) Average $M_2$ in each model updates with and without M2PO masking on Qwen-2.5-32B, showing that masking effectively suppresses spikes and stabilizes the $M_2$ throughout training. (b) Average KL divergence in each model updates on Qwen-2.5-32B under different methods.

**Commonly clipped tokens in GRPO.** Figure 9 shows the specific tokens that are most frequently clipped by $\epsilon$-clipping. The word cloud of commonly clipped tokens is highly aligned with high-entropy tokens shown in (Wang et al., 2025a): these tokens are not random or unimportant, but rather belong to the most semantically and structurally critical elements in reasoning traces. Many of them (e.g., First, simplify, determine, To def, Thus, verify, break) are precisely the high-entropy "pivotal tokens" that initiate, connect, or conclude key reasoning steps. Others (e.g., assistant, user, code markers like ### or $$) serve as structural anchors in the dialogue or mathematical formatting. This observation aligns with Figure 4b: as the importance weight ratio $|r - 1|$ grows, clipped tokens tend to exhibit higher entropy.

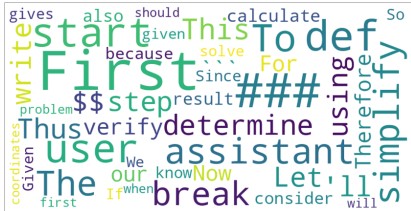

Figure 9: Word clouds of frequently clipped tokens with $\epsilon$ clipping.

## 7 CONCLUSION

In this work, we investigated why off-policy RL for LLMs often fails under stale data and uncovered the *prosperity-before-collapse* phenomenon: training without a trust region initially outperforms standard methods, showing that stale data can be as informative as on-policy trajectories, but eventually collapses due to instability. Motivated by this observation, we proposed **M2PO**, which constrains the second moment of importance weights to provide a variance-sensitive and stable trust region. This design suppresses extreme outliers while preserving informative high-entropy tokens, enabling stable training that matches on-policy performance even under extreme staleness. Extensive experiments further demonstrate that M2PO significantly reduces clipping and is highly insensitive to its threshold, highlighting its practicality and scalability for efficient RL with LLMs.

ACKNOWLEDGEMENT

We would like to thank Cheng Luo, Xinyu Yang, Ranajoy Sadhukhan, Xuesheng Liu, Yongji Wu for providing us constructive feedback on our paper and computing resources of NVIDIA. This work was partially supported by Google Research Award, Google ML & System Junior Faculty Award, Amazon Research Award, Fireworks AI, Intel, Li Auto, Moffett AI, and CMU CyLab Seed funding. This material is also based upon work supported by the National Science Foundation under Grant No. 2504353 and IARPA. Any opinions, findings, and conclusions or recommendations expressed are those of the authors and do not necessarily reflect the views of the National Science Foundation.

## ETHICS STATEMENT

This work focuses on developing reinforcement learning algorithms for large language models. Our research does not involve human subjects, personally identifiable information, or sensitive data. All datasets used are publicly available and widely adopted in the community. We acknowledge that more capable LLMs may have potential societal impacts, including misuse for generating misleading or harmful content. To mitigate these risks, our study is confined to controlled academic settings, and our primary goal is to improve the stability and efficiency of training methods.

## REPRODUCIBILITY STATEMENT

All implementation details, hyperparameters, and experimental setups are thoroughly documented in the paper and appendix, providing sufficient information for independent reproduction of our results. Our code is publicly available at `https://github.com/Infini-AI-Lab/M2PO/`.

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

## A  OVERVIEW

In this appendix, we provide additional details to complement the main text. Appendix B describes the experimental setup in full, including datasets, models, and training hyperparameters. Appendix D presents theoretical proofs supporting our method and analysis.

## B  DETAILED EXPERIMENTAL SETTING

**Models & Datasets.** To verify the effectiveness of our method, we extensively evaluate M2PO on six models from four model series: Qwen2.5-Math-7B (Yang et al., 2024), Llama-3.2-3B-Instruct (Dubey et al., 2024), Qwen3-Base-1.7B/4B/8B (Yang et al., 2025), and Qwen2.5-32B (Yang et al., 2024). For Qwen2.5-Math-7B, we use a context length of 4k, which is the maximum for this series. while for all other models the context length is set to 16k. For training, we adopt the Deep-ScaleR (Luo et al., 2025b) math dataset.

**Training.** Our method is implemented based on verl (Sheng et al., 2024) pipeline and uses vLLM (Kwon et al., 2023) for rollout. We use a mix of H100 and H200 servers for training, depending on resource availability. We set the rollout temperature to 1 for vLLM (Kwon et al., 2023). The training batch size is set to 256 prompts, and the mini-batch size to 64 prompts. We sample 8 responses per prompt. We train all models for 1000 steps, and we optimize the actor model using the AdamW (Loshchilov & Hutter, 2019) optimizer with a constant learning rate of 1e-6. We use $\beta_1 = 0.9$, $\beta_2 = 0.999$, and apply a weight decay of $0.01$. We use the following question template to prompt the LLM. For reward assignment, we give a score of 0.1 for successfully extracting an answer and a score of 1.0 if the extracted answer is correct. Similar to (Yu et al., 2025), we remove the KL-divergence term. The optimization is performed on the parameters of the actor module wrapped with Fully Sharded Data Parallel (FSDP) (Zhao et al., 2023) for efficient distributed training. We set the M2PO threshold to 0.04 for all training runs.

**Evaluation.** For benchmark datasets, we use eight widely used complex mathematical reasoning benchmarks to evaluate the performance of trained models: Math500 (Hendrycks et al., 2021; Lightman et al., 2023), AIME24/25 (Art of Problem Solving, 2024a), AMC23/24 (Art of Problem Solving, 2024b), Minerva Math (Lewkowycz et al., 2022), Gaokao (Zhang et al., 2023), Olympiad Bench (He et al., 2024). Same as the training setting, For Qwen2.5-Math-7B models, we use 4k as the context length. For other models, we set the context length to 16k. Similar to (Wang et al., 2025b; Zheng et al., 2025b), we evaluate models on those benchmarks every 50 steps and report the performance of the checkpoint that obtains the best average performance on eight benchmarks. We evaluate all models with temperature $= 1$. For AIME24/25, we report the $pass@1(avg@16)$, for other benchmarks, we report the $pass@1(avg@4)$.

**Baselines.** For all baseline methods, we adopt exactly the same training setup and evaluation protocol to ensure a fair comparison. For trust-region settings, we follow the recommended hyperparameters reported in the original papers whenever available. Specifically, for GSPO, we use $3e{-}4$ and $4e{-}4$ as the lower and upper bounds. For CISPO, we use a clipping threshold of $0.2$, as the original paper does not specify a particular value; we also experimented with a looser bound, but found that it makes training significantly more prone to collapse. For AREAL, we use a clipping threshold of $0.2$. For GPPO, we set the lower and upper clipping bounds to $0.2$ and $0.28$, respectively. TOPR does not contain any trust-region hyperparameters.

> **Question Template**
>
> Please solve the following math problem: {{Question Description}}. The assistant first thinks about the reasoning process step by step and then provides the user with the answer. Return the final answer in \boxed{} tags, for example \boxed{1}. Let's solve this step by step.

## C  THE USE OF LARGE LANGUAGE MODELS (LLMS)

We used large language models (LLMs) as general-purpose assistants in two limited ways: (1) for writing polish, including improving grammar, readability, and presentation of the manuscript, and

(2) as code assistants (e.g., Cursor, GitHub Copilot) to accelerate routine coding tasks such as debugging syntax errors and refactoring simple functions. LLMs were not used for research ideation, algorithm design, experimental analysis, or drawing conclusions. All conceptual and scientific contributions are entirely the work of the authors.

# D    THEORETICAL PROOF

**Proof of Theorem 1.**

*Proof.* Let $z = \log r$, so that $r = e^Z$. Since $1/R < r \le R$, we have $|z| \le \log R$.

For $z \ge 0$, observe that

$$\frac{e^z - 1}{z} = \int_0^1 e^{tz}\, dt \ \le \ \int_0^1 e^z\, dt = e^z,$$

which implies $(e^z - 1)^2 \le (ze^z)^2 = z^2 e^{2z}$.

For $z \le 0$, set $u = -z \ge 0$. Then

$$(e^z - 1)^2 = (1 - e^{-u})^2 \ \le \ u^2 = z^2 \ \le \ z^2 e^{2|z|}.$$

Combining both cases, for all $z \in \mathbb{R}$ we obtain

$$(e^z - 1)^2 \le z^2 e^{2|z|}.$$

Substituting $Z = \log r$, this yields

$$(r - 1)^2 \le (\log r)^2 e^{2|\log r|} \ \le \ R^2 (\log r)^2.$$

Taking expectation under $\pi_{\text{behav}}$ gives

$$\chi^2(\pi_{\text{new}} \parallel \pi_{\text{behav}}) = \mathbb{E}_{\pi_{\text{behav}}}[(r - 1)^2] \le R^2\, \mathbb{E}_{\pi_{\text{behav}}}[(\log r)^2] = R^2 M_2.$$

$\square$

# E    THEORETICAL INSIGHTS ON PROSPERITY BEFORE COLLAPSE AND M2PO

In this section, we analyze why the unclipped objective is unbiased, while the $\varepsilon$-clipped PPO objective introduces bias by construction. This explains the observed *prosperity-before-collapse* phenomenon.

**Unclipped gradient is unbiased.** Let the importance ratio be

$$r_t(\theta) = \frac{\pi_\theta(a_t \mid s_t)}{\pi_{\text{behav}}(a_t \mid s_t)}.$$

The true policy gradient under off-policy sampling from $\pi_{\text{behav}}$ is

$$\nabla_\theta J(\theta) = \mathbb{E}_{t \sim \pi_{\text{behav}}} \left[ r_t(\theta) A_t \nabla_\theta \log \pi_\theta(a_t \mid s_t) \right].$$

The surrogate objective without clipping is

$$L_{\text{unclip}}(\theta) = \mathbb{E}_t \left[ r_t(\theta) A_t \right],$$

whose gradient matches the true policy gradient,

$$\nabla_\theta L_{\text{unclip}}(\theta) = \mathbb{E}_t [r_t(\theta) A_t \nabla_\theta \log \pi_\theta(a_t \mid s_t)],$$

and is therefore strictly unbiased:

$$\mathbb{E}[\nabla_\theta L_{\text{unclip}}(\theta)] = \nabla_\theta J(\theta).$$

**PPO $\varepsilon$-clipping introduces bias.** The clipped surrogate is

$$L_{\text{clip}}(\theta) = \mathbb{E}_t[\min(r_t(\theta) A_t, \ \text{clip}(r_t(\theta), 1 - \varepsilon, 1 + \varepsilon) A_t)].$$

Consider the set of samples where $(A_t > 0,\, r_t(\theta) > 1 + \varepsilon)$. For these samples PPO replaces $r_t(\theta)$ with a constant $(1 + \varepsilon)$:

$$\min(r_t A_t, (1 + \varepsilon) A_t) = (1 + \varepsilon) A_t,$$

whose gradient is zero:

$$\nabla_\theta \big[ (1 + \varepsilon) A_t \big] = 0.$$

However, the true gradient contribution from these same samples is

$$\nabla_\theta (r_t A_t) = r_t(\theta) A_t \nabla_\theta \log \pi_\theta(a_t \mid s_t) \neq 0.$$

Thus the gradient contribution over this region is removed. Let $\Omega_+ \cup \Omega_-$ denote the set of clipped samples. Then,

$$\nabla_\theta L_{\text{clip}}(\theta) = \mathbb{E}_{t \notin (\Omega_+ \cup \Omega_-)} [r_t A_t \nabla_\theta \log \pi_\theta],$$

whereas the true gradient is

$$\nabla_\theta J(\theta) = \mathbb{E}_t [r_t A_t \nabla_\theta \log \pi_\theta].$$

We denote by $\Omega_+$ and $\Omega_-$ the sets of timesteps where $\varepsilon$-clipping becomes active:

$$\Omega_+ = \{t \mid A_t > 0,\, r_t(\theta) > 1 + \varepsilon\},$$
$$\Omega_- = \{t \mid A_t < 0,\, r_t(\theta) < 1 - \varepsilon\}.$$

Then the difference introduced by clipping is

$$\Delta g(\theta) = -\mathbb{E}_{t \in (\Omega_+ \cup \Omega_-)} [r_t(\theta) A_t \nabla_\theta \log \pi_\theta(a_t \mid s_t)] \neq 0,$$

which shows that PPO's clipped gradient is *biased by construction* whenever clipping occurs.

Combining these properties yields the observed **prosperity-before-collapse phenomenon**: When the clipping is removed, the surrogate gradient becomes the unbiased policy gradient, enabling faster early-stage improvement. However, without a trust region, the importance ratios under off-policy rollouts quickly become heavy-tailed, causing the gradient variance to grow rapidly, eventually leading to training collapse.

The reason M2PO achieves a better trade-off between performance and stability is that its masking strategy is more adaptive. By using the second moment to quantify batch-level distribution shift, M2PO automatically adjusts the effective trust region: when variance is small, it keeps the clipping region wide; when variance grows, it tightens the region accordingly. As a result, the measure of clipped samples, $|\Omega_+ \cup \Omega_-|$, is significantly reduced compared to fixed $\varepsilon$-clipping, leading to a smaller bias while still preventing variance explosion.

## F  DISCUSSION AND ANALYSIS

### F.1  WHY DOES TOLERANCE TO HIGH STALENESS MATTER FOR RL ON LLMS?

High staleness tolerance is both practical and increasingly unavoidable for future RL for LLMs and for scalable RL system designs. Here are two key reasons:

1. High staleness tolerance enables training on more complex tasks. Complex tasks can substantially increase rollout latency. This is especially true for coding and agent environments, where **tool calls** and **reward computation** can be expensive, as both may require executing external programs, interacting with third-party APIs, or running large numbers of test cases to verify correctness.

In realistic settings, such as SWE-bench Jimenez et al. (2023) with OpenHands Wang et al. (2024), rollout latency can become extremely large. A single run, even with a small batch size, the end-to-end inference time (including tool call and code execution) can exceed 100 minutes, with more than 80 iterations with the environment. For reward calculation, verifying correctness against tens or hundreds of test cases also requires repeatedly compiling and running programs, adding another 5–20 minutes of reward computation depending on code and test-case complexity.

As a result, the combined rollout + reward latency can easily reach 120 minutes or more. In contrast, policy updates in LLM RL, especially with sufficient parallelism, can be as fast as 60 seconds per update. Even without any additional system-induced delays, this already corresponds to up to  120 model updates of staleness, and the staleness can be even larger for more complex tasks.

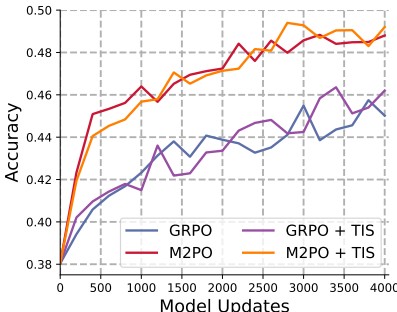

Figure 10: Combining TIS with GRPO and M2PO on Qwen2.5-Math-7B with $s = 256$. Combined with TIS, M2PO shows a slight performance improvement, as TIS better mitigates the distribution gap between FSDP and VLLM. However, TIS alone cannot address the off-policy caused by staleness.

To enable future RL to support increasingly complex LLM agents, especially those involving frequent tool calls and substantial code execution, training under large staleness is practically necessary.

2. High staleness tolerance enables flexible and scalable system design. Modern LLM RL training systems increasingly adopt decoupled and asynchronous designs to improve throughput and reduce cost. In such architectures, the ability to tolerate high staleness directly enables more flexible and scalable RL pipelines for LLMs Fu et al. (2025); Zhu et al. (2025); Wu et al. (2025); Yan et al. (2025), like *heterogeneous computing and disaggregated computing systems*.

1) Heterogeneous Computing. As demonstrated in a recent work Yan et al. (2025), offloading rollouts to slower but cost-efficient machines can substantially reduce the overall training expense. However, this will significantly increase the rollout latency, leading to larger staleness.

2) Disaggregated Computing. In large-scale RL training systems, rollout generation and policy updates can run on physically separated clusters or even across different data centers. Network bandwidth and latency can become a huge bottleneck: synchronizing model weights and transferring rollout trajectories can further amplify staleness as the system scales.

To sum up, these factors make high staleness a realistic and sometimes unavoidable operating regime in modern RL for LLMs. Tolerance to high staleness is therefore crucial not only for cost-efficient and flexible system design, but also for enabling RL on complex long-latency tasks without sacrificing throughput.

Motivated by this, we use a clean and controlled stale-k abstraction to isolate the effect of data staleness on RL algorithms. Within this controlled but representative setup, we show the limitation of GRPO under high staleness, while M2PO remains stable and approaches on-policy performance. We believe this work provides algorithmic understandings of RL under high staleness and lays the foundation for future system-level integration.

## F.2 HOW IS M2PO RELATED TO TIS (TRUNCATED IMPORTANCE SAMPLING)?

Another off-policy RL technique, TIS (Liu et al., 2025a), addresses a different source of off-policy drift: the mismatch between the inference engine used for rollouts (e.g., vLLM, SGLang, or quantized engines) and the training engine (e.g., FSDP or Megatron). Despite that the weights are exactly the same, the engine difference on training and inference can introduce sampling differences and cause potential mismatch issues. This form of off-policy discrepancy is orthogonal to the source we study: **data staleness**.

More specifically, the optimization objective for TIS in (Liu et al., 2025a) is:

$$\mathbb{E}_{\pi_{\text{vllm}}(\theta_{\text{behav}})}\left[\min\left(\frac{\pi_{\text{fsdp}}(a, \theta_{\text{behav}})}{\pi_{\text{vllm}}(a, \theta_{\text{behav}})}, C\right) \cdot \nabla_\theta \min\left(\frac{\pi_{\text{fsdp}}(a, \theta)}{\pi_{\text{fsdp}}(a, \theta_{\text{behav}})}\hat{A}, \text{clip}\left(\frac{\pi_{\text{fsdp}}(a, \theta)}{\pi_{\text{fsdp}}(a, \theta_{\text{behav}})}, 1 - \epsilon, 1 + \epsilon\right)\hat{A}\right)\right]$$

We can see that the TIS objective introduces a truncated (or scaled) importance ratio to prevent excessively large ratios and stabilize training under inference-engine mismatch. However, for the

off-policy drift caused by staleness, TIS still relies on a fixed $\epsilon$-based clipping region to form a trust region. In contrast, the primary goal of M2PO is to provide an adaptive, variance-sensitive mechanism to determine the trust region based on the second-moment statistics of importance ratios, without relying on a hand-chosen $\epsilon$. In this sense, TIS and M2PO are complementary: TIS mitigates inference-engine mismatch; M2PO stabilizes training under stale data and large-degree off-policy drift. Here is a formula combining TIS and M2PO:

$$\mathcal{J}_{\text{M2PO}}(\theta) = \frac{1}{\sum_{i=1}^{G} |o_i|} \sum_{i=1}^{G} \sum_{t=1}^{|o_i|} \min\left( \frac{\pi_{\theta_{\text{behav}},\text{fsdp}}(o_i|q)}{\pi_{\theta_{\text{behav}},\text{vllm}}(o_i|q)}, C \right) \boldsymbol{M}_{i,t} \frac{\pi_{\theta,\text{fsdp}}(o_i|q)}{\pi_{\theta_{\text{behav}},\text{fsdp}}(o_i|q)} A_{i,t}, \quad (6)$$

where $\boldsymbol{M}_{i,t}$ is the mask generated by Algorithm 1.

In Figure 10, we evaluate the combination of TIS with both GRPO and M2PO on Qwen2.5-Math-7B under a high-staleness setting ($s = 256$). When integrated with TIS, M2PO achieves a slight performance improvement, as TIS helps reduce the distribution gap between the FSDP training engine and the VLLM rollout engine. (That said, the gain is modest because both training and rollout are performed in FP16, which keeps the engine-induced distribution shift relatively small.) However, TIS alone is insufficient to address the distribution shift introduced by large staleness.

### F.3 COMPARE M2PO WITH ASYMMETRIC CLIPPING

We have conducted an additional experiment using the asymmetric clipping ratio (0.2, 0.28) used in DAPO for our setting and report the results in Figure 11. Despite the asymmetric clipping ratio improving the GRPO performance, it still underperforms M2PO.

The fundamental distinction between M2PO and asymmetric clipping is twofold:

1) **M2PO adaptively determines the trust region based on the actual distribution shift of each training batch.** The second-moment statistic automatically tightens the trust region when the shift becomes large, and relaxes it when the shift is small, thereby preserving more informative gradients. This per-batch adaptivity cannot be reproduced by any static $\epsilon$-clipping threshold: a fixed upper/lower bound does not respond to the evolving off-policy drift during training.

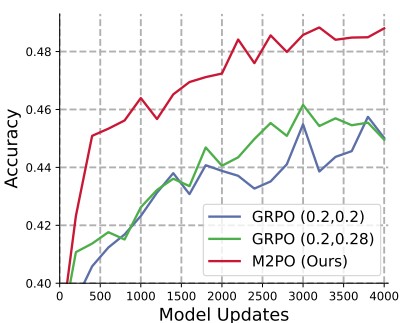

Figure 11: The performanace comparison between M2PO and GRPO on coding tasks.

2) **Tuning clipping ratios is highly brittle and fundamentally impractical for different settings.** The optimal asymmetric bounds vary across model sizes, staleness levels, and even across different phases of training. The tuning complexity of asymmetric clipping becomes even higher because both upper and lower bounds must be selected jointly (quadratic complexity). In contrast, the M2PO threshold directly reflects the current batch's distribution shift, which is the reason that the M2PO threshold is not sensitive, and we can use the same threshold in all our experiments.

We believe that these two reasons illustrate why simply adjusting clipping boundaries is not sufficient for stabilizing stale off-policy training and why the community continues to develop principled trust-region algorithms rather than relying on hand-tuned clipping thresholds.

### F.4 M2PO COMPUTATION COST ANALYSIS

Although we use per-token second-moment ($M_2$) estimates to compute the batch-level $M_2$ statistic, **we do not require logits or probabilities over the entire vocabulary**. Instead, $M_2$ is computed only from the **sampled token** at each position (i.e., a single-sample Monte Carlo estimate). Thus, $M_2$ estimation adds no additional forward-pass computation and no additional tensor storage, avoiding the $O(\text{VocabularySize})$ cost.

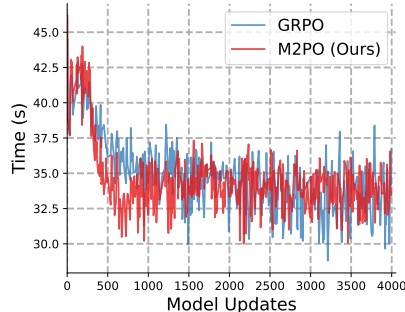

Figure 12: Training time (i.e., update policy) comparison between GRPO and M2PO. M2PO masking does not introduce a noticeable overhead in training stage.

Table 2: Comparison of computation time between GRPO and M2PO. Loss computation contributes a negligible portion of the total training time.

|  | **Loss Compute Time (s)** | **Other Training Time (s)** |
| --- | --- | --- |
| **GRPO** | 0.038 (0.11%) | 34.86 (99.89%) |
| **M2PO (Ours)** | 0.065 (0.19%) | 34.43 (99.81%) |

Besides, for the masking procedure, our implementation sorts tokens by their $M_2$ values once and iteratively masks those with the largest $M_2$ until the remaining tokens' average $M_2$ falls below the threshold $\tau_{M_2}$. This procedure has a time complexity of $O(N \log N)$, where $N$ is the number of tokens in a batch. In practice, $N$ is typically a few thousand, and this sorting step is negligible compared to the forward/backward passes of large language models. As we illustrated in Figure 12, GRPO and M2PO show a similar training time, which indicates that M2PO masking does not introduce a noticeable overhead in the training stage.

We further provide a breakdown of the loss-computation overhead (including masking computation) during the entire training process. Table 2 below reports the average loss-computation time and the remaining training time (including forward and backward passes) for RL training on Qwen2.5-Math-7B. As shown, M2PO introduces a slightly higher computational cost than GRPO due to the additional sorting required to select tokens with the highest M2 values. However, this overhead is extremely small relative to the full training step and is far from the bottleneck. For instance, M2PO spends only 0.19% of total time on loss computation and does not slow down the overall training process.

Overall, both $M2$ estimation and the M2PO masking procedure incur **minimal computational and memory overhead**, and do not slow down training or increase storage requirements in our experiments.

### F.5    How is M2PO related to KL constraint/loss?

The KL-divergence loss calculated in the Deepseek R1 (DeepSeek-AI et al., 2025) is a low-variance approximation of the true KL divergence between the old and new policies (Schulman, 2015), which can partially mitigate off-policy drift.

In our early exploration, we also experimented with adding a KL loss. While this regularizer does enhance stability, we found that a trust region is still required to prevent collapse. Once a trust region is introduced, however, we immediately encounter the same challenge discussed in Q3: *how to select an adaptive and effective trust region boundary that balances stability and performance*. Simply tuning a KL penalty or fixed KL target does not adequately address the dynamic distribution shift introduced by stale data.

Moreover, the KL constraint and M2PO are complementary. The KL penalty controls global deviation between the new and old policies, whereas M2PO adaptively masks high-shift tokens based on per-batch second-moment statistics. Combining the two is straightforward and can potentially further improve stability. In this work, we intentionally did not include a KL penalty in our main

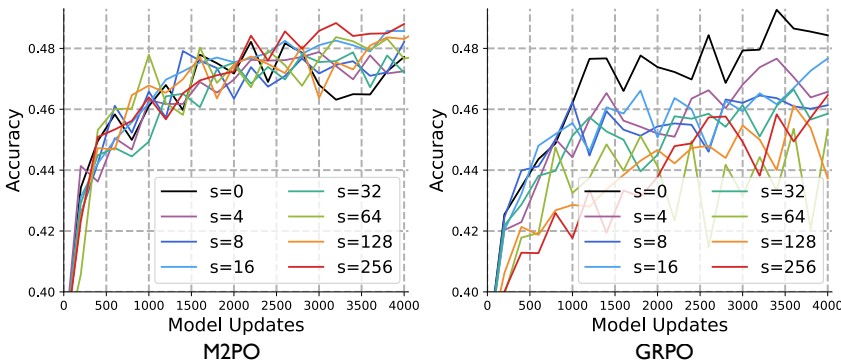

Figure 13: The performance of M2PO and GRPO under different staleness on Qwen2.5-Math-7B.

experiments. Our goal is to isolate and understand the role of the trust-region mechanism itself under stale off-policy training. We will explore the combination of KL regularization and M2PO as a promising future direction.

# G  ADDITIONAL EXPERIMENTS

## G.1  M2PO UNDER DIFFERENT STALENESS

In Table 1, we compare the performance between M2PO and other baselines under $s = 256$. To further study the robustness of M2PO across different staleness levels, we train Qwen2.5-Math-7B with M2PO under $s = 0, 32, 64, 128, 256$, as shown in Figure 13. We observe that larger staleness slightly slows down the initial accuracy gain during the very early phase of training, reflecting the expected delay when using outdated rollouts. However, this effect quickly diminishes: all curves steadily improve and converge to nearly the same accuracy. Notably, even under extreme staleness ($s = 256$), M2PO achieves performance on par with the on-policy case ($s = 0$), without signs of collapse or degradation. These results highlight that M2PO effectively preserves the learning signal contained in stale data, enabling stable training and strong generalization across a wide range of staleness values.

## G.2  TRAINING COLLAPSE ANALYSIS

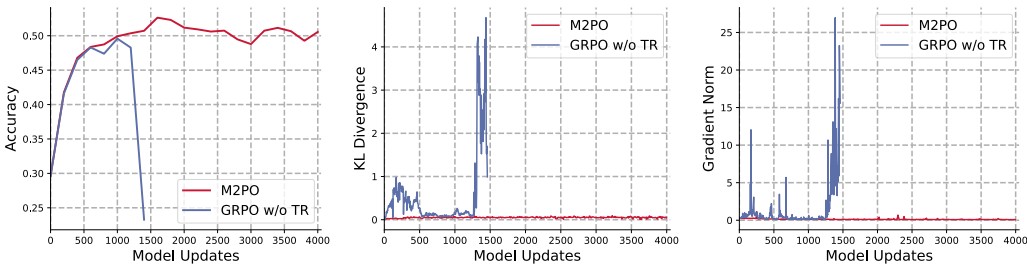

Figure 14: Qwen2.5-32B training dynamics comparison on M2PO and collapse training of GRPO without a trust region. **Left:** Test accuracy across eight math benchmarks. **Middle:** KL divergence between current policy and reference policy. **Right:** Graident norm.

In our evaluation, we deem training to have collapsed when we observe a pronounced drop in test accuracy accompanied by abnormalities in stability metrics such as KL divergence and gradient norms. In practice, accuracy degradation is typically paired with sharp KL spikes and gradient-norm blow-ups. As shown in Figure 14, test accuracy drops at the same time that KL divergence and gradient norms exhibit consistent spikes.

