# OpenReview forum: "Prosperity before Collapse: How Far Can Off-Policy RL Reach with Stale Data on LLMs?"
_ICLR.cc/2026/Conference — ICLR 2026 Poster_

### Official Review · Reviewer_8kZV · 2025-10-28

**Soundness:** 3
**Presentation:** 3
**Contribution:** 3
**Rating:** 4
**Confidence:** 3

**Summary:**

This paper introduces M2PO (Second-Moment Trust Policy Optimization), a novel off-policy reinforcement learning algorithm designed to address the challenges of training large language models (LLMs) with stale data. Current on-policy RL methods, while stable, are inefficient and unscalable due to their reliance on fresh rollouts. Off-policy methods, which decouple rollout generation from training, often suffer from performance degradation or collapse when data staleness is high.

The authors identify a "prosperity-before-collapse" phenomenon where stale data can initially be as informative as on-policy data if properly exploited, but existing methods like GRPO fail due to excessive clipping of informative high-entropy tokens. M2PO tackles this by constraining the second moment of importance weights, effectively suppressing extreme outliers while preserving valuable informative updates. This approach dramatically reduces token clipping under high staleness, leading to stable optimization.

The paper demonstrates that M2PO achieves performance comparable to on-policy GRPO even with data stale by at least 256 model updates, across six models (1.7B to 32B) and eight math reasoning benchmarks. M2PO is shown to be robust to its threshold hyperparameter, simplifying its practical application. This work highlights the potential of stale data in RL for LLMs and provides a more stable and efficient training solution.

**Strengths:**

This paper presents several strengths that would be beneficial for reviewers to consider:

- **Addresses a Critical Problem in LLM Training:** The paper tackles the significant challenge of efficiently training LLMs with reinforcement learning. The reliance of traditional RL on fresh data (on-policy) is a major bottleneck for scalability, and M2PO offers a practical solution to leverage stale data.
- **Identifies a Novel Phenomenon:** The "prosperity-before-collapse" phenomenon is an intriguing observation that sheds light on the potential of stale data. This insight forms the foundation of their proposed method.
- **Proposes a Well-Motivated Solution (M2PO):** M2PO directly addresses the identified problem of existing methods inappropriately clipping informative tokens. By constraining the second moment of importance weights, the algorithm effectively differentiates between noisy outliers and valuable high-entropy tokens, leading to more stable and efficient learning.
- **Strong Empirical Evidence:** The extensive evaluation across six different LLMs (ranging from 1.7B to 32B parameters) and eight math reasoning benchmarks provides compelling evidence of M2PO's effectiveness. The consistent performance parity with on-policy methods, even under extreme data staleness, is a key highlight.

**Weaknesses:**

1. Computational Overhead of M2PO and Masking: The computational overhead associated with calculating KL divergence, which can involve operations at the scale of $VocabularySize \times TokenNumber$, and the iterative nature of the masking algorithm (Algorithm 1) are not thoroughly detailed. More explanation is needed regarding the practical implications of these computations, including tensor storage requirements and time complexity, especially when dealing with very large language models and extensive vocabularies.

2. Limited Evaluation on Small-Degree Staleness and Practical Relevance of Extreme Staleness: The paper primarily focuses on evaluating M2PO under large-degree staleness (e.g., 256 model updates), while several real-world RL training setups and recent works (e.g., DAPO with 16 gradient updates per rollout, or AREAL uses even smaller degree of staleness) demonstrate that smaller degrees of staleness (e.g., 4, 8, or 16) can be sufficient for maximizing efficiency. This raises two important questions:

   - How does M2PO perform under these more commonly encountered small-degree staleness settings? A comparison in these scenarios would provide a more complete picture of its practical utility.
   - What is the necessity of large-degree staleness (e.g., 256) in real-world RL training for LLMs? Further exploration of this aspect could clarify the specific contexts in which such high staleness is beneficial or required.

**Questions:**

In my understanding, the experiments only involves using stable data (i.e., data sample with old behavior models) but doesn't involve any reusing of sampled data (e.g., one sample prompt can be used more than once in gradient updating), am I right?

---

> ### Author Response · Authors · 2025-11-21
> **Response to Reviewer 4 (8kZV) (1/3)**
>
> We thank the reviewer for addressing a critical problem in RL for LLMs, identifying a novel phenomenon, and proposing a well-motivated solution with strong empirical performance. We now address the reviewer’s specific questions below:
>
>
> > **Q1:** Computational Overhead of M2PO and Masking: The computational overhead associated with calculating KL divergence, which can involve operations at the scale of $VocabularySize \times TokenNumber$, and the iterative nature of the masking algorithm (Algorithm 1) are not thoroughly detailed. More explanation is needed regarding the practical implications of these computations, including tensor storage requirements and time complexity, especially when dealing with very large language models and extensive vocabularies.
>
> **A1:** We thank the reviewer for raising this important point regarding the computational overhead of M2PO. We have included a training time comparison on the training stage, which includes the mask calculation, between M2PO and GRPO in $\textcolor{blue}{\text{Figure 12 (Appendix F.4)}}$. The results show that M2PO masking computation does not introduce noticeable overhead.
>
>
>
> **M2PO adds negligible memory or runtime cost.**
>
> 1. Although we use per-token second-moment ($M_2$) estimates to compute the batch-level $M_2$ statistic, **we do not require logits or probabilities over the entire vocabulary**. Instead, similar to GRPO, $M_2$ is computed only from the **sampled token** at each position(i.e., a single-sample Monte Carlo estimate):
> $M_2 = [\log \frac{\pi_{\theta}(a_i \mid s_i)}{\pi_{\text{behav}}(a_i \mid s_i)}]^2$, where $\pi_{\theta}(a_i \mid s_i)$ and $\pi_{\text{behav}}(a_i \mid s_i)$ are the probability of the sampled token. Thus, $M_2$ estimation needs no additional tensor storage, **avoiding the $O(\text{VocabularySize})$ cost** that the reviewer was concerned about.
>
> 2. For the masking procedure, our implementation sorts tokens by their $M2$ values once and iteratively masks those with the largest $M2$ until the remaining tokens’ average $M2$ falls below the threshold $\tau_{\text{M2}}$. This procedure has a time complexity of **$O(N \log N)$**, where $N$ is the number of tokens in a batch. This sorting step is negligible compared to the forward/backward passes of large language models. We have updated Algorithm 1 to include more details.
>
>
>
> As shown in the training-stage time comparison in $\textcolor{blue}{\text{ Figure 12 (Appendix F.4)}}$, the training stage, including the masking computation, has a comparable runtime for M2PO and GRPO, indicating that M2PO introduces only negligible additional overhead.
>
>
> We thank the reviewer again for the comments. We have updated $\textcolor{blue}{\text{Algorithm 1 in Section 5.2}}$ to make it clearer and included the cost analysis results and discussion in $\textcolor{blue}{\text{Appendix F.4}}$ in the revised paper to clarify this point.

---

> > ### Author Response · Authors · 2025-11-21
> > **Response to Reviewer 4 (8kZV) (2/3)**
> >
> > > **Q2:** What is the necessity of large-degree staleness (e.g., 256) in real-world RL training for LLMs? Further exploration of this aspect could clarify the specific contexts in which such high staleness is beneficial or required.
> >
> > **A2:**  We are happy to clarify why high staleness tolerance is both practical and increasingly unavoidable for future RL for LLMs and for scalable RL system designs. Here are two key reasons:
> > ### 1. High staleness tolerance enables training on more complex tasks.
> >
> > Complex tasks can substantially increase rollout latency. This is especially true for coding and agent environments, where **tool calls** and **reward computation** can be expensive, as both may require executing external programs, interacting with third-party APIs, or running large numbers of test cases to verify correctness.
> >
> > In realistic settings, such as SWE-bench[1] with OpenHands[2], rollout latency can become extremely large. A single run, even with a small batch size, the end-to-end inference time (including tool call and code execution) can exceed 100 minutes, with more than 80 iterations with the environment.  For reward calculation, verifying correctness against tens or hundreds of test cases also requires repeatedly compiling and running programs, adding another 5–20 minutes of reward computation depending on code and test-case complexity.
> >
> > As a result, the combined rollout + reward latency can easily reach 120 minutes or more. In contrast, policy updates in LLM RL, especially with sufficient parallelism, can be as fast as 60 seconds per update. Even without any additional system-induced delays, this already corresponds to up to ~120 model updates of staleness, and the staleness can be even larger for more complex tasks.
> >
> > To enable future RL to support increasingly complex LLM agents, especially those involving frequent tool calls and substantial code execution, training under large staleness is practically necessary.
> >
> > ### 2. High staleness tolerance enables flexible and scalable system design.
> >
> > Modern LLM RL training systems increasingly adopt decoupled and asynchronous designs to improve throughput and reduce cost. In such architectures, the ability to tolerate high staleness directly enables more flexible and scalable RL pipelines for LLMs [3-6], like *heterogeneous computing and disaggregated computing systems*.
> >
> > 1) Heterogeneous Computing. As demonstrated in a recent work [6], offloading rollouts to slower but cost-efficient machines can substantially reduce the overall training expense. However, this will significantly increase the rollout latency, leading to larger staleness.
> >
> > 2) Disaggregated Computing. In large-scale RL training systems, rollout generation and policy updates can run on physically separated clusters or even across different data centers. Network bandwidth and latency can become a huge bottleneck: synchronizing model weights and transferring rollout trajectories can further amplify staleness as the system scales.
> >
> > ---
> >
> > To sum up, these factors make high staleness a realistic and sometimes unavoidable operating regime in modern RL for LLMs. Tolerance to high staleness is therefore crucial not only for cost-efficient and flexible system design, but also for enabling RL on complex long-latency tasks without sacrificing throughput.
> >
> >
> > Motivated by this, in this paper, we use a clean and controlled stale-k abstraction to isolate the effect of data staleness on RL algorithms. Within this controlled but representative setup, we show the limitation of GRPO under high staleness, while M2PO remains stable and approaches on-policy performance. We believe this work provides algorithmic understandings of RL under high staleness and lays the foundation for future system-level integration.

---

> ### Author Response · Authors · 2025-11-21
> **Response to Reviewer 4 (8kZV) (3/3)**
>
> > **Q3:** How does M2PO perform under these more commonly encountered small-degree staleness settings? A comparison in these scenarios would provide a more complete picture of its practical utility.
>
> **A3:**  We thank the reviewer for the insightful suggestion. We have added an evaluation of M2PO under different staleness settings (s=4,8,16,32,64,128) in $\textcolor{blue}{\text{Figure 15 (Appendix G.3)}}$. The results show that **M2PO maintains performance comparable to the non-stale setting (s = 0) across a wide range of staleness values**, indicating that our method is robust to both mild and extreme staleness. In contrast, as we discussed in Section 3.2, GRPO usually experiences a performance drop when training with stale data. This demonstrates a better practical applicability of M2PO in real-world RL training setups where staleness is typically moderate. In this paper, we use 256 staleness setting to serve as a more challenging upper-bound setting to evaluate the robustness limits of M2PO.
>
> We have included the results and discussion in $\textcolor{blue}{\text{Figure 15 (Appendix G.3)}}$ in the revised paper.
>
>
> > **Q4:** In my understanding, the experiments only involves using stable data (i.e., data sample with old behavior models) but doesn't involve any reusing of sampled data (e.g., one sample prompt can be used more than once in gradient updating), am I right?
>
> **A4:** Yes, your understanding is correct. Each data point will be used only once in the entire training. We have clarified this point and updated the description in $\textcolor{blue}{\text{Section 3.2}}$  of the revised paper to make it more explicit.
>
> ---
>
>
> Thanks for the attentive reading of the manuscript and constructive feedback. We have incorporated these changes into our revised version. We hope our response addresses all the concerns and that the reviewer will consider raising the rating accordingly. We are more than glad to answer any further questions.
>
>
> ---
>
>
> [1] Jimenez, Carlos E., John Yang, Alexander Wettig, Shunyu Yao, Kexin Pei, Ofir Press, and Karthik Narasimhan. "Swe-bench: Can language models resolve real-world github issues?." arXiv preprint arXiv:2310.06770 (2023).
>
> [2] Wang, Xingyao, Boxuan Li, Yufan Song, Frank F. Xu, Xiangru Tang, Mingchen Zhuge, Jiayi Pan et al. "Openhands: An open platform for ai software developers as generalist agents." arXiv preprint arXiv:2407.16741 (2024).
>
> [3] Fu, Wei, Jiaxuan Gao, Xujie Shen, Chen Zhu, Zhiyu Mei, Chuyi He, Shusheng Xu et al. "AReaL: A Large-Scale Asynchronous Reinforcement Learning System for Language Reasoning." arXiv preprint arXiv:2505.24298 (2025).
>
> [4] Zhu, Zilin, Chengxing Xie, Xin Lv, and slime Contributors. “slime: A Large-Scale LLM Post-Training Framework for RL Scaling.” GitHub repository, 2025. https://github.com/THUDM/slime.
>
> [5] Wu, Yongji, Xueshen Liu, Haizhong Zheng, Juncheng Gu, Beidi Chen, Z. Morley Mao, Arvind Krishnamurthy, and Ion Stoica. "RLBoost: Harvesting Preemptible Resources for Cost-Efficient Reinforcement Learning on LLMs." arXiv preprint arXiv:2510.19225 (2025).
>
> [6] Yan, Ran, Youhe Jiang, Tianyuan Wu, Jiaxuan Gao, Zhiyu Mei, Wei Fu, Haohui Mai, Wei Wang, Yi Wu, and Binhang Yuan. "AReaL-Hex: Accommodating Asynchronous RL Training over Heterogeneous GPUs." arXiv preprint arXiv:2511.00796 (2025).

---

> > ### Author Response · Authors · 2025-11-26
> > **Follow-up Discussion and Experiments**
> >
> > Hi Reviewer 8kZV,
> >
> > We thank you again for the time and effort you dedicated to reviewing our paper, as well as for the insightful comments and suggestions that helped us further improve the work.
> >
> > We hope that our original responses are helpful to address your concerns. To better answer your questions, beyond the discussion and experiments included in our original response, we have now incorporated further experiments that provide additional evidence supporting our claims:
> >
> >
> > **Detailed breakdown of Computation Cost of M2PO masking (for Q1)**:  We further provide a breakdown of the loss-computation overhead (including masking computation) during the entire training process. The table below reports the average loss-computation time and the remaining training time (including forward and backward passes) for RL training on Qwen2.5-Math-7B.
> >
> > |             | Loss Compute Time (s) | Other Training Time (s) |
> > |-------------|-----------------------|-------------------------|
> > | GRPO        |         0.038 (0.11%) |         34.86s (99.89%) |
> > | M2PO (Ours) |        0.065 (0.19%) |         34.43s (99.81%) |
> >
> > As shown, M2PO introduces a slightly higher computational cost than GRPO due to the additional sorting required to select tokens with the highest M2 values. However, this overhead is extremely small relative to the full training step and is **far from the bottleneck**. For instance, M2PO spends only **0.19%** of total time on loss computation and does not slow down the overall training process.
> >
> > **More Comparison under Different Staleness (for Q4)**: We further evaluated the performance of GRPO under varying staleness levels and updated the results in $\textcolor{blue}{\text{Figure 15 (Appendix G.3)}}$ in the revised paper for a better comparison to the proposed M2PO. The comparison shows that M2PO remains robust across a broad range of staleness values and achieves performance comparable to the on-policy setting, whereas GRPO exhibits a clear performance drop as staleness increases.
> >
> >
> > ---
> >
> > Thanks again for the comments and suggestions in your review, which are very beneficial to help improve our paper. We have included all your suggestions in our revised paper. We hope our response addresses all the concerns, and we are more than glad to answer any further questions!

---

### Official Review · Reviewer_D8e7 · 2025-10-31

**Soundness:** 3
**Presentation:** 3
**Contribution:** 2
**Rating:** 6
**Confidence:** 4

**Summary:**

This paper addresses the challenge of performance degradation in Reinforcement Learning (RL) for Large Language Models (LLMs) when trained on stale, off-policy data. The authors identify a "prosperity before collapse" phenomenon, where training without a trust region initially performs well but eventually fails.

To harness this potential while ensuring stability, they propose M2PO (Second-Moment Trust Policy Optimization). This algorithm constrains the second moment of importance weights, which selectively masks only extreme, high-variance tokens while preserving informative updates from stale data.

Extensive experiments show that M2PO enables stable training even with data stale by 256 model updates, matching on-policy performance and significantly outperforming existing methods across various model scales and reasoning benchmarks.

**Strengths:**

- The paper presents an interesting observation that training without clipping on stale data can initially outperform clipped training, highlighting a "prosperity-before-collapse" phenomenon.

- The proposed method, M2PO, is intuitive and easy to follow. It selectively excludes tokens that cause extreme increases in the batch-level second-moment metric during loss propagation.

**Weaknesses:**

See Questions.

**Questions:**

- Regarding the "stale-k" setup: Is the model using a policy from k training steps earlier to generate the current training batch, or is it directly using data collected k steps ago? If it’s the former, I suggest using the term "stale model" for clarity.

- Why is such a large staleness (e.g., s=256) used? While off-policy training is an important challenge in RL post-training, techniques like Truncated Importance Sampling (TIS) [1] can help maintain on-policy behavior. Could the authors compare M2PO with TIS under similar settings?

- Including an ablation study with different staleness levels for M2PO would help better understand its robustness and performance across varying off-policy gaps.

[1] Your Efficient RL Framework Secretly Brings You Off-Policy RL Training. https://fengyao.notion.site/off-policy-rl

---

> ### Author Response · Authors · 2025-11-21
> **Response to Reviewer 3 (D8e7) (1/3)**
>
> We appreciate that Reviewer D8e7 highlighted both the novelty of our “prosperity-before-collapse’’ observation and the clarity of the M2PO method. We are glad the reviewer found the phenomenon insightful and the design of M2PO intuitive. We now address the reviewer’s specific questions below:
>
> > **Q1:** Regarding the "stale-k" setup: Is the model using a policy from k training steps earlier to generate the current training batch, or is it directly using data collected k steps ago? If it’s the former, I suggest using the term "stale model" for clarity.
>
> **A1:**  Thank you for the question. In our “stale-k” setup, the training batch at step $t$ is collected $k$ updates earlier, but we implement this by storing the generated data rather than storing old checkpoints. Concretely, at update step $t-k$ we use the current policy to generate rollouts and place them into a buffer, and after $k$ further updates, these trajectories are consumed once for training at step $t$. Each sampled trajectory is used exactly once and is never reused across multiple updates.
>
> We have revised $\textcolor{blue}{\text{Section 3.2}}$ in the paper to clarify this implementation and the meaning of the “stale-k” terminology.
>
>
> > **Q2:** Why is such a large staleness (e.g., s=256) used?
>
> **A2:**  We are happy to clarify why high staleness tolerance is both practical and increasingly unavoidable for future RL for LLMs and for scalable RL system designs. Here are two key reasons:
> ### 1. High staleness tolerance enables training on more complex tasks.
>
> Complex tasks can substantially increase rollout latency. This is especially true for coding and agent environments, where **tool calls** and **reward computation** can be expensive, as both may require executing external programs, interacting with third-party APIs, or running large numbers of test cases to verify correctness.
>
> In realistic settings, such as SWE-bench[1] with OpenHands[2], rollout latency can become extremely large. A single run, even with a small batch size, the end-to-end inference time (including tool call and code execution) can exceed 100 minutes, with more than 80 iterations with the environment.  For reward calculation, verifying correctness against tens or hundreds of test cases also requires repeatedly compiling and running programs, adding another 5–20 minutes of reward computation depending on code and test-case complexity.
>
> As a result, the combined rollout + reward latency can easily reach 120 minutes or more. In contrast, policy updates in LLM RL, especially with sufficient parallelism, can be as fast as 60 seconds per update. Even without any additional system-induced delays, this already corresponds to up to ~120 model updates of staleness, and the staleness can be even larger for more complex tasks.
>
> To enable future RL to support increasingly complex LLM agents, especially those involving frequent tool calls and substantial code execution, training under large staleness is practically necessary.
>
> ### 2. High staleness tolerance enables flexible and scalable system design.
>
> Modern LLM RL training systems increasingly adopt decoupled and asynchronous designs to improve throughput and reduce cost. In such architectures, the ability to tolerate high staleness directly enables more flexible and scalable RL pipelines for LLMs [3-6], like *heterogeneous computing and disaggregated computing systems*.
>
> 1) Heterogeneous Computing. As demonstrated in a recent work [6], offloading rollouts to slower but cost-efficient machines can substantially reduce the overall training expense. However, this will significantly increase the rollout latency, leading to larger staleness.
>
> 2) Disaggregated Computing. In large-scale RL training systems, rollout generation and policy updates can run on physically separated clusters or even across different data centers. Network bandwidth and latency can become a huge bottleneck: synchronizing model weights and transferring rollout trajectories can further amplify staleness as the system scales.
>
> ---
>
> To sum up, these factors make high staleness a realistic and sometimes unavoidable operating regime in modern RL for LLMs. Tolerance to high staleness is therefore crucial not only for cost-efficient and flexible system design, but also for enabling RL on complex long-latency tasks without sacrificing throughput.
>
>
> Motivated by this, in this paper, we use a clean and controlled stale-k abstraction to isolate the effect of data staleness on RL algorithms. Within this controlled but representative setup, we show the limitation of GRPO under high staleness, while M2PO remains stable and approaches on-policy performance. We believe this work provides algorithmic understandings of RL under high staleness and lays the foundation for future system-level integration.

---

> ### Author Response · Authors · 2025-11-21
> **Response to Reviewer 3 (D8e7) (2/3)**
>
> > **Q3:** While off-policy training is an important challenge in RL post-training, techniques like Truncated Importance Sampling (TIS) [7] can help maintain on-policy behavior. Could the authors compare M2PO with TIS under similar settings?
>
> **A3:** We thank the reviewer for bringing TIS (Truncated Importance Sampling)[7] for discussion. We would like to clarify that TIS and M2PO are complementary rather than competing approaches, and TIS alone cannot handle off-policy issues caused by staleness. We have additionally evaluated a combined variant that applies TIS on top of M2PO, and the results are reported in $\textcolor{blue}{\text{Figure 10 (Appendix F.2)}}$.
>
> TIS addresses a different source of off-policy drift: **the mismatch between the inference engine used for rollouts (e.g., vLLM, SGLang, or quantized engines) and the training engine (e.g., FSDP or Megatron)**. Despite that the weights are exactly the same, the engine difference on training and inference can introduce sampling differences and cause potential mismatch issues. This form of off-policy discrepancy is orthogonal to the source we study: **data staleness**.
>
> More specifically, the optimization objective for TIS is:
>
> $$E_{\pi_{\mathrm{vllm}}(\theta_{\mathrm{behav}})}\left[\min\left(\frac{\pi_{\mathrm{fsdp}}(a,\theta_{\mathrm{behav}})}{\pi_{\mathrm{vllm}}(a,\theta_{\mathrm{behav}})},C\right)\cdot\nabla_\theta\min\left(\frac{\pi_{\mathrm{fsdp}}(a,\theta)}{\pi_{\mathrm{fsdp}}(a,\theta_{\mathrm{behav}})}\hat A,\operatorname{clip}\left(\frac{\pi_{\mathrm{fsdp}}(a,\theta)}{\pi_{\mathrm{fsdp}}(a,\theta_{\mathrm{behav}})},1-\epsilon,1+\epsilon\right)\hat A\right)\right]$$
>
>
> We can see that the TIS objective introduces a truncated (or scaled) importance ratio to stabilize training. However, for the off-policy drift caused by staleness, TIS still relies on a fixed $\epsilon$-based clipping region to form a trust region.
> In contrast, the primary goal of M2PO is to provide an adaptive, variance-sensitive mechanism to determine the trust region based on the second-moment statistics of importance ratios, without relying on a hand-chosen $\epsilon$. In this sense, TIS and M2PO are complementary:
> 1. TIS mitigates inference-engine mismatch.
> 2. M2PO stabilizes training under stale data and large-degree off-policy drift.
>
> Here is a formula combining TIS and M2PO:
> $$J_{M2PO}(\theta)=\frac{1}{\sum_{i=1}^G |o_i|}\sum_{i=1}^G\sum_{t=1}^{|o_i|}\min\left(\frac{\pi_{\theta_{behav},fsdp}(o_i\mid q)}{\pi_{\theta_{behav},vllm}(o_i\mid q)},C\right) M_{i,t}\frac{\pi_{\theta,fsdp}(o_i\mid q)}{\pi_{\theta_{behav},fsdp}(o_i\mid q)}A_{i,t},$$
>
> where $M_{i,t}$ is mask gotten from Algorithm 1 in our paper.
>
> We have combined TIS with GRPO and M2PO and trained on Qwen2.5-Math-7B. The results are reported in $\textcolor{blue}{\text{Figure 10 (Appendix F.2)}}$: when integrated with TIS, M2PO achieves a slight performance improvement, as TIS helps reduce the distribution gap between the FSDP training engine and the VLLM rollout engine. However, TIS alone is insufficient to address the distribution shift introduced by large staleness, as the $\epsilon$-clipping still masks out important tokens for training.
>
> We have also added a new subsection ($\textcolor{blue}{\text{F.2}}$) in the Appendix to discuss the conceptual relationship between TIS and M2PO and illustrate how the two methods can be combined.

---

> > ### Author Response · Authors · 2025-11-21
> > **Response to Reviewer 3 (D8e7) (3/3)**
> >
> > > **Q4:** Including an ablation study with different staleness levels for M2PO would help better understand its robustness and performance across varying off-policy gaps.
> >
> > **A4:** We thank the reviewer for the insightful suggestion. We have added an evaluation of M2PO under different staleness settings (s=4,8,16,32,64,128) in $\textcolor{blue}{\text{Figure 15 (Appendix G.3)}}$. The results show that **M2PO maintains performance comparable to the non-stale setting (s = 0) across a wide range of staleness values**, indicating that our method is robust to both mild and extreme staleness. In contrast, as we discussed in Section 3.2, GRPO usually experiences a performance drop when training with stale data. This demonstrates a better practical applicability of M2PO in real-world RL training setups where staleness is typically moderate. In the paper, we use 256 staleness setting to serve as a more challenging upper-bound setting to evaluate the robustness limits of M2PO.
> >
> > We have included the results and discussion in $\textcolor{blue}{\text{Figure 15 (Appendix G.3)}}$ in the revised paper.
> >
> > ---
> >
> > Thanks for the attentive reading of the manuscript and constructive feedback. We have incorporated these changes into our revised version. We hope our response addresses all the concerns and that the reviewer will consider raising the rating accordingly. We are more than glad to answer any further questions.
> >
> > ---
> >
> > [1] Jimenez, Carlos E., John Yang, Alexander Wettig, Shunyu Yao, Kexin Pei, Ofir Press, and Karthik Narasimhan. "Swe-bench: Can language models resolve real-world github issues?." arXiv preprint arXiv:2310.06770 (2023).
> >
> > [2] Wang, Xingyao, Boxuan Li, Yufan Song, Frank F. Xu, Xiangru Tang, Mingchen Zhuge, Jiayi Pan et al. "Openhands: An open platform for ai software developers as generalist agents." arXiv preprint arXiv:2407.16741 (2024).
> >
> > [3] Fu, Wei, Jiaxuan Gao, Xujie Shen, Chen Zhu, Zhiyu Mei, Chuyi He, Shusheng Xu et al. "AReaL: A Large-Scale Asynchronous Reinforcement Learning System for Language Reasoning." arXiv preprint arXiv:2505.24298 (2025).
> >
> > [4] Zhu, Zilin, Chengxing Xie, Xin Lv, and slime Contributors. “slime: A Large-Scale LLM Post-Training Framework for RL Scaling.” GitHub repository, 2025. https://github.com/THUDM/slime.
> >
> > [5] Wu, Yongji, Xueshen Liu, Haizhong Zheng, Juncheng Gu, Beidi Chen, Z. Morley Mao, Arvind Krishnamurthy, and Ion Stoica. "RLBoost: Harvesting Preemptible Resources for Cost-Efficient Reinforcement Learning on LLMs." arXiv preprint arXiv:2510.19225 (2025).
> >
> > [6] Yan, Ran, Youhe Jiang, Tianyuan Wu, Jiaxuan Gao, Zhiyu Mei, Wei Fu, Haohui Mai, Wei Wang, Yi Wu, and Binhang Yuan. "AReaL-Hex: Accommodating Asynchronous RL Training over Heterogeneous GPUs." arXiv preprint arXiv:2511.00796 (2025).
> >
> > [7] Your Efficient RL Framework Secretly Brings You Off-Policy RL Training. https://fengyao.notion.site/off-policy-rl

---

> > > ### Author Response · Authors · 2025-11-26
> > > **Follow-up Discussion and Experiments**
> > >
> > > Hi Reviewer D8e7,
> > >
> > > We thank you again for the time and effort you dedicated to reviewing our paper, as well as for the insightful comments and suggestions that helped us further improve the work.
> > >
> > > We hope that our original responses are helpful to address your concerns. To better answer your questions, beyond the discussion and experiments included in our original response, we have now incorporated further experiments that provide additional evidence supporting our claims:
> > >
> > > **More Comparison under Different Staleness (for Q4)**: We further evaluated the performance of GRPO under varying staleness levels and updated the results in $\textcolor{blue}{\text{Figure 15 (Appendix G.3)}}$ in the revised paper for a better comparison to the proposed M2PO. The comparison shows that M2PO remains robust across a broad range of staleness values and achieves performance comparable to the on-policy setting, whereas GRPO exhibits a clear performance drop as staleness increases.
> > >
> > > ---
> > >
> > > Thanks again for the comments and suggestions in your review, which are very beneficial to help improve our paper. We have included all your suggestions in our revised paper. We hope our response addresses all the concerns, and we are more than glad to answer any further questions!

---

### Official Review · Reviewer_vuPK · 2025-10-31

**Soundness:** 3
**Presentation:** 3
**Contribution:** 2
**Rating:** 6
**Confidence:** 2

**Summary:**

This paper investigates a critical challenge in scaling RL for LLMs: performance degradation when training with stale data in off-policy settings. The authors identify a "prosperity-before-collapse" phenomenon, where training without a trust region initially performs well on stale data before eventual collapse, suggesting that stale data contains valuable but hard-to-harness information. To address this, they propose M2PO, a novel policy optimization algorithm that constrains the second moment $M_2$ of the importance weights. M2PO employs an adaptive masking strategy to suppress only extreme, high-variance tokens, thereby preserving informative updates. The method is evaluated extensively across 6 model scales (1.7B to 32B) and 8 math reasoning benchmarks, demonstrating that M2PO achieves performance comparable to on-policy baselines even with data stale by 256 model updates, while significantly reducing the token clipping ratio and maintaining training stability.

**Strengths:**

- The proposed M2PO algorithm is built upon a well-motivated and sound theoretical insight. It identifies the limitations of KL divergence in measuring distribution shift — specifically its insensitivity to outliers due to cancellation effects — and introduces the second moment ($M_2$) as a superior alternative. The $M_2$ metric is both an effective measure of distribution shift and inherently sensitive to the high-variance tokens that are most likely to destabilize training.

- The fact that a single threshold hyperparameter $\tau_{M_2} = 0.04$ works robustly across all model sizes and tasks is a major practical advantage, greatly enhancing the method's usability.

- The empirical evaluation is extensive and compelling. The consistent success of M2PO across six model families and scales under extreme staleness $s=256$ provides strong evidence for its robustness and scalability, demonstrating performance on par with on-policy baselines.

**Weaknesses:**

- While the paper compellingly demonstrates M2PO's superiority under extreme staleness $s=256$, it remains unclear if this advantage holds under more common, lower-staleness regimes (e.g., $s=8$ or $s=16$). A critical comparison in a moderate-staleness setting would be highly informative.

- The empirical evaluation is currently confined to mathematical reasoning benchmarks. While this domain is a valid testbed, it features verifiable rewards and potentially specific characteristics in its token-wise importance ratio $r$ distribution. The claims regarding M2PO's general superiority and the universal robustness of its threshold $\pi_{M_2}$ would be significantly strengthened by evaluation on other domains, such as code generation.

**Questions:**

- The paper convincingly shows the superiority of M2PO over standard GRPO with symmetric clipping. However, a natural alternative is to use GRPO with asymmetric clipping bounds. Could the authors discuss the theoretical and practical relationship between M2PO and such an asymmetric GRPO variant? Why is M2PO's adaptive masking fundamentally different from simply tuning the clipping boundaries?

- We note that the KL divergence calculation in DeepSeek-R1's GRPO implementation, $\frac{\pi_{new}}{\pi_{behav}} - \log \frac{\pi_{new}}{\pi_{behav}} - 1$, also possesses the property of being more sensitive to large ratios compared to the standard KL. Given this, what is the distinct advantage of M2PO over such an improved, large-ratio-sensitive KL divergence used within the existing GRPO framework?

---

> ### Author Response · Authors · 2025-11-21
> **Response to Reviewer 2 (vuPK) (1/3)**
>
> We are glad that Reviewer vuPK found M2PO to be well-motivated, grounded in sound theoretical insight, equipped with a robust and stable thresholding mechanism, and demonstrating strong empirical performance across a wide range of settings. We now address the reviewer’s specific questions below:
>
> > **Q1.1:** While the paper compellingly demonstrates M2PO's superiority under extreme staleness, it remains unclear if this advantage holds under more common, lower-staleness regimes (e.g., s=8 or s=16 ). A critical comparison in a moderate-staleness setting would be highly informative.
>
> **A1.1:**  We thank the reviewer for the insightful suggestion. We have added an evaluation of M2PO under different staleness settings (s=4,8,16,32,64,128) in $\textcolor{blue}{\text{Figure 15 (Appendix G.3)}}$. The results show that **M2PO maintains performance comparable to the non-stale setting (s = 0) across a wide range of staleness values**, indicating that our method is robust to both mild and extreme staleness. In contrast, as we discussed in Section 3.2, GRPO usually experiences a performance drop when training with stale data. This demonstrates a better practical applicability of M2PO in real-world RL training setups where staleness is typically moderate. In the paper, we use 256 staleness setting to serve as a more challenging upper-bound setting to evaluate the robustness limits of M2PO.
>
> We have included the results and discussion in $\textcolor{blue}{\text{Figure 15 (Appendix G.3)}}$ in the revised paper.
>
> > **Q1.2:** Why does tolerance to high staleness also matter?
>
> **A1.2:**  Moreover, we would like to further clarify why high staleness tolerance is both practical and increasingly unavoidable for future RL for LLMs and for scalable RL system designs. Here are two key reasons:
>
> ### 1. High staleness tolerance enables training on more complex tasks.
>
> Complex tasks can substantially increase rollout latency. This is especially true for coding and agent environments, where **tool calls** and **reward computation** can be expensive, as both may require executing external programs, interacting with third-party APIs, or running large numbers of test cases to verify correctness.
>
> In realistic settings, such as SWE-bench[1] with OpenHands[2], rollout latency can become extremely large. A single run, even with a small batch size, the end-to-end inference time (including tool call and code execution) can exceed 100 minutes, with more than 80 iterations with the environment.  For reward calculation, verifying correctness against tens or hundreds of test cases also requires repeatedly compiling and running programs, adding another 5–20 minutes of reward computation depending on code and test-case complexity.
>
> As a result, the combined rollout + reward latency can easily reach 120 minutes or more. In contrast, policy updates in LLM RL, especially with sufficient parallelism, can be as fast as 60 seconds per update. Even without any additional system-induced delays, this already corresponds to up to ~120 model updates of staleness, and the staleness can be even larger for more complex tasks.
>
> To enable future RL to support increasingly complex LLM agents, especially those involving frequent tool calls and substantial code execution, training under large staleness is practically necessary.
>
> ### 2. High staleness tolerance enables flexible and scalable system design.
>
> Modern LLM RL training systems increasingly adopt decoupled and asynchronous designs to improve throughput and reduce cost. In such architectures, the ability to tolerate high staleness directly enables more flexible and scalable RL pipelines for LLMs [3-6], like *heterogeneous computing and disaggregated computing systems*.
>
> 1) Heterogeneous Computing. As demonstrated in a recent work [6], offloading rollouts to slower but cost-efficient machines can substantially reduce the overall training expense. However, this will significantly increase the rollout latency, leading to larger staleness.
>
> 2) Disaggregated Computing. In large-scale RL training systems, rollout generation and policy updates can run on physically separated clusters or even across different data centers. Network bandwidth and latency can become a huge bottleneck: synchronizing model weights and transferring rollout trajectories can further amplify staleness as the system scales.
>
> ---
>
> To sum up, these factors make high staleness a realistic and sometimes unavoidable operating regime in modern RL for LLMs. Tolerance to high staleness is therefore crucial not only for cost-efficient and flexible system design, but also for enabling RL on complex long-latency tasks without sacrificing throughput.

---

> ### Author Response · Authors · 2025-11-21
> **Response to Reviewer 2 (vuPK) (2/3)**
>
> > **Q2:**  The claims regarding M2PO's general superiority and the universal robustness of its threshold $\tau_{M_2}$  would be significantly strengthened by evaluation on other domains, such as code generation.
>
> **A2:** Yes, **M2PO can still stay effective beyond math reasoning**. To further evaluate the generalizability of M2PO, we have launched an RL training on the code generation domain. Specifically, we trained DeepSeek-R1-Distill-Qwen-1.5B on the code_contests dataset[7] and evaluated the models on the LiveCodeBench[8] benchmark. For M2PO, we still use the same $M_2$ threshold, i.e., $\tau_{M_2} = 0.04$.
>
> As RL for code generation is typically very slow, since each rollout must execute test cases to verify correctness and compute final rewards, we have so far completed 2,000 model updates. We report the interim results in $\textcolor{blue}{\text{Figure 13 (Appendix G.1)}}$. Although training is not yet fully finished, we already observe that **M2PO outperforms GRPO under stale-data training and matches the performance of on-policy GRPO**.
>
>
> We will update the complete results once the full training run is done.
>
> > **Q3:** The paper convincingly shows the superiority of M2PO over standard GRPO with symmetric clipping. However, a natural alternative is to use GRPO with asymmetric clipping bounds. Could the authors discuss the theoretical and practical relationship between M2PO and such an asymmetric GRPO variant? Why is M2PO's adaptive masking fundamentally different from simply tuning the clipping boundaries?
>
> **A3:** We have conducted an additional experiment using the asymmetric clipping ratio used in DAPO for our setting and report the results in $\textcolor{blue}{\text{Figure 11 (Appendix F.3)}}$ in the revised paper. We can see that, with this asymmetric setting, GRPO still underperforms M2PO.
>
> The **fundamental distinction** between M2PO and asymmetric clipping is twofold:
>
>  1) **M2PO adaptively determines the trust region based on the actual distribution shift of each training batch.** The second-moment statistic automatically tightens the trust region when the shift becomes large, and relaxes it when the shift is small, thereby preserving more informative gradients. This per-batch adaptivity cannot be reproduced by any static $\epsilon$-clipping threshold: a fixed upper/lower bound does not respond to the evolving off-policy drift during training.
>
>  2) **Tuning clipping ratios is highly brittle and fundamentally impractical for different settings.** The optimal asymmetric bounds vary across model sizes, staleness levels, and even across different phases of training. The tuning complexity of asymmetric clipping becomes even higher because both upper and lower bounds must be selected jointly (quadratic complexity). In contrast, the M2PO threshold directly reflects the current batch’s distribution shift, which is the reason that the M2PO threshold is not sensitive, and we can use the same threshold in all our experiments.
>
> We believe that these two reasons illustrate why simply adjusting clipping boundaries is not sufficient for stabilizing stale off-policy training and why the community continues to develop principled trust-region algorithms rather than relying on hand-tuned clipping thresholds. We have added the asymmetric-clipping experiments and a detailed discussion in $\textcolor{blue}{\text{Appendix F.3}}$.

---

> ### Author Response · Authors · 2025-11-21
> **Response to Reviewer 2 (vuPK) (3/3)**
>
> > **Q4:**  We note that the KL divergence calculation in DeepSeek-R1's GRPO implementation, , also possesses the property of being more sensitive to large ratios compared to the standard KL. Given this, what is the distinct advantage of M2PO over such an improved, large-ratio-sensitive KL divergence used within the existing GRPO framework?
>
>
> **A4:** The KL-divergence calculation in the Deepseek R1 paper is a low-variance approximation of the true KL divergence between the old and new policies (as noted in Schulman’s blog [9]), which can partially mitigate off-policy drift.
>
> In our early exploration, we also experimented with adding a KL loss. While this regularizer does enhance stability, we found that a trust region is still required to prevent collapse. Once a trust region is introduced, however, we immediately encounter the same challenge discussed in Q3: *how to select an adaptive and effective trust region boundary that balances stability and performance*. Simply tuning a KL penalty or fixed KL target does not adequately address the dynamic distribution shift introduced by stale data.
>
> Moreover, the KL constraint and M2PO are complementary. The KL penalty controls global deviation between the new and old policies, whereas M2PO adaptively masks high-shift tokens based on per-batch second-moment statistics. Combining the two is straightforward and can potentially further improve stability.
>
> In this work, we intentionally did not include a KL penalty in our main experiments. Our goal is to isolate and understand the role of the trust-region mechanism itself under stale off-policy training. We will explore the combination of KL regularization and M2PO as a promising future direction.
>
> We have added this clarification and additional discussion to $\textcolor{blue}{\text{Appendix F.5}}$ in the revised paper.
>
> ---
>
> Thanks for the attentive reading of the manuscript and constructive feedback. We have incorporated these changes into our revised version. We hope our response addresses all the concerns and that the reviewer will consider raising the rating accordingly. We are more than glad to answer any further questions.
>
> ---
>
>
>
> [1] Jimenez, Carlos E., John Yang, Alexander Wettig, Shunyu Yao, Kexin Pei, Ofir Press, and Karthik Narasimhan. "Swe-bench: Can language models resolve real-world github issues?." arXiv preprint arXiv:2310.06770 (2023).
>
> [2] Wang, Xingyao, Boxuan Li, Yufan Song, Frank F. Xu, Xiangru Tang, Mingchen Zhuge, Jiayi Pan et al. "Openhands: An open platform for ai software developers as generalist agents." arXiv preprint arXiv:2407.16741 (2024).
>
> [3] Fu, Wei, Jiaxuan Gao, Xujie Shen, Chen Zhu, Zhiyu Mei, Chuyi He, Shusheng Xu et al. "AReaL: A Large-Scale Asynchronous Reinforcement Learning System for Language Reasoning." arXiv preprint arXiv:2505.24298 (2025).
>
> [4] Zhu, Zilin, Chengxing Xie, Xin Lv, and slime Contributors. “slime: A Large-Scale LLM Post-Training Framework for RL Scaling.” GitHub repository, 2025. https://github.com/THUDM/slime.
>
> [5] Wu, Yongji, Xueshen Liu, Haizhong Zheng, Juncheng Gu, Beidi Chen, Z. Morley Mao, Arvind Krishnamurthy, and Ion Stoica. "RLBoost: Harvesting Preemptible Resources for Cost-Efficient Reinforcement Learning on LLMs." arXiv preprint arXiv:2510.19225 (2025).
>
> [6] Yan, Ran, Youhe Jiang, Tianyuan Wu, Jiaxuan Gao, Zhiyu Mei, Wei Fu, Haohui Mai, Wei Wang, Yi Wu, and Binhang Yuan. "AReaL-Hex: Accommodating Asynchronous RL Training over Heterogeneous GPUs." arXiv preprint arXiv:2511.00796 (2025).
>
> [7] Li, Yujia, David Choi, Junyoung Chung, Nate Kushman, Julian Schrittwieser, Rémi Leblond, Tom Eccles et al. "Competition-level code generation with alphacode." Science 378, no. 6624 (2022): 1092-1097.
>
> [8] Jain, Naman, King Han, Alex Gu, Wen-Ding Li, Fanjia Yan, Tianjun Zhang, Sida Wang, Armando Solar-Lezama, Koushik Sen, and Ion Stoica. "Livecodebench: Holistic and contamination free evaluation of large language models for code." arXiv preprint arXiv:2403.07974 (2024).
>
> [9] http://joschu.net/blog/kl-approx.html. Approximating KL Divergence

---

> > ### Author Response · Authors · 2025-11-26
> > **Follow-up Discussion and Experiments**
> >
> > Hi Reviewer vuPK,
> >
> > We thank you again for the time and effort you dedicated to reviewing our paper, as well as for the insightful comments and suggestions that helped us further improve the work.
> >
> > We hope that our original responses are helpful to address your concerns. To better answer your questions, beyond the discussion and experiments included in our original response, we have now incorporated further experiments that provide additional evidence supporting our claims:
> >
> > **More Comparison under Different Staleness (for Q1)**: We further evaluated the performance of GRPO under varying staleness levels and updated the results in $\textcolor{blue}{\text{Figure 15 (Appendix G.3)}}$ in the revised paper for a better comparison to the proposed M2PO. The comparison shows that M2PO remains robust across a broad range of staleness values and achieves performance comparable to the on-policy setting, whereas GRPO exhibits a clear performance drop as staleness increases.
> >
> > **Completed Training Results on Coding Tasks (for Q2)**:  We have completed the training on the coding tasks with 4000 model updates and updated the complete training curve in $\textcolor{blue}{\text{Figure 13 (Appendix G.1)}}$. From the evaluation results, we observe that **M2PO outperforms GRPO under stale-data training and matches the performance of on-policy GRPO**.
> >
> > ---
> >
> > Thanks again for the comments and suggestions in your review, which are very beneficial to help improve our paper. We have included all your suggestions in our revised paper. We hope our response addresses all the concerns, and we are more than glad to answer any further questions!

---

### Official Review · Reviewer_GUjw · 2025-11-03

**Soundness:** 2
**Presentation:** 2
**Contribution:** 2
**Rating:** 4
**Confidence:** 3

**Summary:**

The paper investigates why off‑policy RL for LLMs degrades when trained on stale data (rollouts from earlier policies). It reveals a clear “prosperity‑before‑collapse” phenomenon: removing the trust region yields substantially higher performance than standard GRPO with epsilon‑clipping for a period, sometimes matching on‑policy results, but eventually becomes unstable and collapses. The authors diagnose that GRPO performs poorly under staleness because stale‑data training exhibits a much higher clipping rate, disproportionately affecting informative, high‑entropy tokens. To address this, they propose M2PO, a variance‑sensitive trust region that constrains the second moment of importance weights. M2PO suppresses extreme outliers while preserving signal on high‑entropy tokens, achieves stable training, matches on‑policy performance even under high staleness (e.g., 256 updates on Qwen‑2.5‑32B), dramatically reduces clipping, and is reported to be insensitive to its threshold.

**Strengths:**

+ Clear, timely empirical finding: prosperity‑before‑collapse highlights that stale data can be as informative as on‑policy trajectories, shifting the focus from data quality to algorithm design.
+ Concrete diagnosis of GRPO’s failure mode under staleness: elevated clipping, especially on high‑entropy tokens, plausibly explains lost training signal.
+ M2PO’s second‑moment constraint provides a simple, principled, variance‑sensitive trust region that stabilizes off‑policy training.
+ on Qwen‑2.5‑32B with staleness 256, M2PO matches on‑policy performance while reducing clipping events and avoiding collapse。

**Weaknesses:**

No formal analysis of why the unclipped regime prospers then collapses, or why second‑moment constraints deliver stability.
Incomplete experimental detail: definitions of staleness, collapse criteria, and full baseline configurations are not specified in the provided materials, making external validity hard to assess.
It is unclear how results vary across model sizes, tasks, reward models, and different staleness levels beyond the highlighted case.
Compute matching, training schedules, and hyperparameter tuning parity between on‑policy GRPO, GRPO without trust region, and M2PO are not fully documented.
Direct token‑level gradient/credit assignment analyses are not shown here to firmly establish the masking/clipping mechanism.

**Questions:**

How exactly is “staleness of 256 model updates” defined? Is it the number of policy updates between rollout generation and training, or another measure?
Do you track a policy‑distance metric (e.g., KL between rollout policy and training policy) and how does M2PO perform as this distance changes?
What operational criteria define “collapse”? Reward crash, divergence of losses, KL blow‑up, or mode collapse?
What stability diagnostics did you monitor, and can you show trajectories (e.g., gradient norms, importance‑weight second moments) leading to collapse?
How do the results vary across tasks, reward models, and model sizes? Can you share performance under different staleness levels beyond 256?

---

> ### Author Response · Authors · 2025-11-21
> **Response to Reviewer 1 (GUjw)  (1/3)**
>
> ## Response for Reviewer 1 (#GUjw)
>
> We are glad that Reviewer GUjw found our findings and diagnosis clear and timely, and appreciated the principled design of our method as well as its strong empirical performance under off-policy data induced by staleness. We now address the reviewer’s specific questions below:
>
> > **Q1:**  No formal analysis of why the unclipped regime prospers then collapses, or why second‑moment constraints deliver stability.
>
> **A1:** We have added a more detailed theoretical analysis in $\textcolor{blue}{\text{Appendix E}}$ of the revised paper.
>
> To summarize, standard $\varepsilon$-clipping introduces bias into the policy-gradient update. Specifically, the resulting gradient bias is
> $\Delta g(\theta)= -E_{t\in(\Omega_+ \cup \Omega_-)}\left[r_t(\theta)A_t\nabla_\theta\log\pi_\theta(a_t\mid s_t)\right]\neq 0,$
> where $\Omega_+$ and $\Omega_-$ are clipped token sets for positive and negative examples. Intuitively, clipping forces the update to ignore part of the valid gradient signal, and thus removing the trust region can reduce this bias and improve performance. However, removing clipping altogether leads to substantially higher variance, which in turn causes instability and eventual training collapse, which explains the prosperity before collapse scenario.
>
> M2PO achieves a better trade-off between performance and stability because its masking mechanism is more adaptive. By using the second moment to quantify batch-level distribution shift, M2PO automatically adjusts the effective trust region: when the variance is small, it allows a wider clipping region; when the variance grows, it tightens the region accordingly. This adaptivity significantly reduces the number of clipped samples, $|\Omega_+ \cup \Omega_-|$, compared with fixed $\varepsilon$-clipping, as also evidenced in Figure 7 of the main paper, thereby reducing gradient bias while still maintaining training stability.
>
> A more complete version of the theoretical discussion is provided in $\textcolor{blue}{\text{Appendix E}}$. We hope these analyses address your concerns.

---

> ### Author Response · Authors · 2025-11-21
> **Response to Reviewer 1 (GUjw) (2/3)**
>
> > **Q2:**  [Training Settings Clarification] Incomplete experimental detail: definitions of staleness, collapse criteria, and full baseline configurations are not specified in the provided materials, making external validity hard to assess. Compute matching, training schedules, and hyperparameter tuning parity between on‑policy GRPO, GRPO without trust region, and M2PO are not fully documented.
>
> **A2:** We are happy to clarify the training settings used in this paper and explain why our experiment is a fair comparison.
>
> ### 1) Experimental Setting
> For the experimental setup, all methods were trained under matched compute budgets, identical rollout schedules, identical batch sizes, identical training lengths, and the same inference/training infrastructure, ensuring a fair comparison (all configurations are listed in $\textcolor{blue}{\text{Appendix B}}$). The training configuration used in this work follows the setting adopted by several recent GRPO papers [1,2]. In other words, **these configurations were originally optimized for GRPO, not M2PO. We did not modify or retune any of these settings specifically for M2PO.**
> We have updated $\textcolor{blue}{\text{Appendix B}}$ to include all baseline configurations, training configurations, and hyperparameters.
> ### 2) Definition of Stalness
>
> > How exactly is “staleness of 256 model updates” defined? Is it the number of policy updates between rollout generation and training, or another measure? Do you track a policy‑distance metric (e.g., KL between rollout policy and training policy), and how does M2PO perform as this distance changes?
>
> We are happy to further clarify the meaning of “staleness of 256 model updates.” As described in Section 3.2, staleness refers to the number of policy updates between rollout generation and training. For a given training iteration, all training data are sampled from the policy snapshot taken k updates earlier; these data have not been used in any previous training step and are each used exactly once. We also evaluate M2PO under a range of staleness levels and observe that M2PO remains robust across different degrees of staleness. (Please see Q4 for more details)
>
> We have improved the presentation in $\textcolor{blue}{\text{Section 3.2}}$ to make this definition clearer.
>
> ### 3) Collapse Criteria
>
> >What operational criteria define “collapse”? Reward crash, divergence of losses, KL blow‑up, or mode collapse? What stability diagnostics did you monitor, and can you show trajectories (e.g., gradient norms, importance‑weight second moments) leading to collapse?
>
> In our experiment, we consider training to have collapsed when we observe a significant drop in test accuracy, together with abnormalities in stability metrics such as KL divergence and gradient norms. In practice, the degradation in test accuracy is usually accompanied by KL spikes and gradient-norm blow-ups. Therefore, we believe that our chosen metrics provide a reasonable and reliable criterion for determining training collapse.
>
> We have included a collapse training dynamic (accuracy, KL divergence, and gradient norm dynamics of Qwen2.5-32b model training) in $\textcolor{blue}{\text{Figure 16 in Appendix G.4}}$ to provide more evidence on training collapse. These plots show that KL divergence and gradient norms consistently spike at the same time that test accuracy falls. These plots show that KL divergence and gradient norms consistently spike at the same time that test accuracy falls.
>
> We have incorporated these additional analyses and clarifications in $\textcolor{blue}{\text{ Appendix G.4}}$ in the revised paper.
>
> **To sum up, we believe that these clarifications and diagnostics ensure transparency and demonstrate the fairness and validity of our experimental evaluation.**

---

> > ### Author Response · Authors · 2025-11-21
> > **Response to Reviewer 1 (GUjw)  (3/3)**
> >
> > > **Q3:** Direct token‑level gradient/credit assignment analyses are not shown here to firmly establish the masking/clipping mechanism.
> >
> > **A3:** We thank the reviewer for bringing the token-level gradient/credit-assignment for discussion. Due to the extremely sparse reward structure in RLVR, accurately estimating per-token contributions remains an open challenge. Recent work [3] shows that **high-entropy tokens disproportionately influence training**: using only the top 20% high-entropy tokens (and masking the remaining 80%) can yield performance comparable to using all tokens.
> >
> > In Section 4 of our paper, we observe a closely related phenomenon: **clipped tokens under GRPO are more likely to be high-entropy tokens**. As staleness increases, the distribution shift enlarges, leading to masking out even more of these high-entropy tokens, which directly affects training performance.
> >
> > To further illustrate this, we include a visualization of frequently clipped tokens in $\textcolor{blue}{\text{Figure 14 (Appendix G.2)}}$. Many tokens (e.g., First, simplify, determine, To, def, Thus, verify, break) are exactly the kinds of “pivotal high-entropy tokens” identified in prior work: tokens that initiate, connect, or conclude key reasoning steps (similar to Figure 2b in [3]). This provides complementary evidence that clipped tokens remove critical reasoning signals and explains why excessive clipping leads to performance degradation under large staleness. M2PO mitigates this issue by adaptively adjusting the trust region, allowing more of these high-entropy tokens to contribute to training while maintaining stability.
> > We have added this additional discussion and visualization to $\textcolor{blue}{\text{Appendix G.2}}$ in the revised paper.
> >
> > > **Q4:**  It is unclear how results vary across model sizes, tasks, reward models, and different staleness levels beyond the highlighted case. How do the results vary across tasks, reward models, and model sizes? Can you share performance under different staleness levels beyond 256?
> >
> > **A4:** We thank the reviewer for the insightful suggestion. We have added an evaluation of M2PO under different staleness settings (s=4,8,16,32,64,128) in $\textcolor{blue}{\text{Figure 15 (Appendix G.3)}}$. The results show that **M2PO maintains performance comparable to the non-stale setting (s = 0) across a wide range of staleness values**, indicating that our method is robust to both mild and extreme staleness. In contrast, as we discussed in Section 3.2, GRPO usually experiences a performance drop when training with stale data. This demonstrates a better practical applicability of M2PO in real-world RL training setups where staleness is typically moderate. In the paper, we use 256 staleness setting to serve as a more challenging upper-bound setting to evaluate the robustness limits of M2PO.
> >
> >
> > We have included the results and discussion in $\textcolor{blue}{\text{Figure 15 (Appendix G.3)}}$ in the revised paper.
> >
> >
> > ---
> >
> > Thanks for the attentive reading of the manuscript and constructive feedback. We have incorporated these changes into our revised version. We hope our response addresses all the concerns and that the reviewer will consider raising the rating accordingly. We are more than glad to answer any further questions!
> >
> > ---
> >
> > [1] Zheng, H., Zhou, Y., Bartoldson, B. R., Kailkhura, B., Lai, F., Zhao, J., & Chen, B. (2025). Act Only When It Pays: Efficient Reinforcement Learning for LLM Reasoning via Selective Rollouts. NeurIPS 2025
> >
> > [2] Wang, Yiping, Qing Yang, Zhiyuan Zeng, Liliang Ren, Liyuan Liu, Baolin Peng, Hao Cheng et al. "Reinforcement learning for reasoning in large language models with one training example." NeurIPS 2025
> >
> > [3] Wang, Shenzhi, Le Yu, Chang Gao, Chujie Zheng, Shixuan Liu, Rui Lu, Kai Dang et al. "Beyond the 80/20 rule: High-entropy minority tokens drive effective reinforcement learning for llm reasoning." NeurIPS 2025

---

> > > ### Author Response · Authors · 2025-11-26
> > > **Follow-up Discussion and Experiments**
> > >
> > > Hi Reviewer GUjw,
> > >
> > > We thank you again for the time and effort you dedicated to reviewing our paper, as well as for the insightful comments and suggestions that helped us further improve the work.
> > >
> > > We hope that our original responses are helpful to address your concerns. To better answer your questions, beyond the discussion and experiments included in our original response, we have now incorporated further experiments that provide additional evidence supporting our claims:
> > >
> > > **More Comparison under Different Staleness (for Q4)**: We further evaluated the performance of GRPO under varying staleness levels and updated the results in $\textcolor{blue}{\text{Figure 15 (Appendix G.3)}}$ in the revised paper for a better comparison to the proposed M2PO. The comparison shows that M2PO remains robust across a broad range of staleness values and achieves performance comparable to the on-policy setting, whereas GRPO exhibits a clear performance drop as staleness increases.
> > >
> > > ---
> > >
> > > Thanks again for the comments and suggestions in your review, which are very beneficial to help improve our paper. We have included all your suggestions in our revised paper. We hope our response addresses all the concerns, and we are more than glad to answer any further questions!

---

### Author Response · Authors · 2025-11-21
**General response to all reviewers**

Dear all reviewers,

We sincerely thank all the reviewers for their thoughtful reviews and invaluable feedback. We truly appreciate the time and effort you devoted to carefully reading our paper and providing constructive comments. Your insights have been extremely helpful in strengthening the clarity, rigor, and presentation of our work.

We are grateful that the reviewers recognized several positive aspects of our method M2PO, including: “clear and interesting observations and findings” (All reviewers), “principled and clear method” (GUjw, D8e7, 8kZV), “strong empirical performance” (GUjw, vuPK, 8kZV), “insensitivity to parameter selection” (vuPK), and “study and address a critical problem” (8kZV).


For the comments and concerns raised in the reviews, we have prepared individual, point-by-point responses for each reviewer. We have also revised the manuscript accordingly, and we summarize the key changes below. To facilitate cross-referencing, all modifications are highlighted in blue in the updated version.

Overall, the revised paper includes **6 additional pages** for new discussion and **7 new figures**, reflecting the substantial clarifications, analyses, and improvements inspired by the reviewers’ feedback.


## Paper Revision Summary:

**1. Additional evaluation and analysis:**


1. $\textcolor{blue}{\text{Figure 10 in Appendix F.2}}$: Evaluation on combining TIS with M2PO; (suggested by Reviewer D8e7)

2. $\textcolor{blue}{\text{Figure 11 in Appendix F.3}}$: GRPO with asymmetric clipping. (suggested by Reviewer vuPK)

3. $\textcolor{blue}{\text{Figure 12 in Appendix F.4}}$: Training stage time comparison between GRPO and M2PO. (suggested by Reviewer 8kZV)

4. $\textcolor{blue}{\text{Figure 13 in Appendix G.1}}$: Evaluation on code generation tasks. (suggested by Reviewer vuPK)

5. $\textcolor{blue}{\text{Figure 14 in Appendix G.2}}$: Visualization of commonly clipped tokens; (suggested by Reviewer GUjw)

6. $\textcolor{blue}{\text{Figure 15 in Appendix G.3}}$: M2PO evaluation under different staleness. (suggested by all reviewers)

7. $\textcolor{blue}{\text{Figure 16 in Appendix G.4}}$: Training collapse analysis; (suggested by Reviewer GUjw)


**2. Writing:**

1. Update $\textcolor{blue}{\text{Section 1}}$ (Introduction)  to better clarify why tolerance to high staleness is practical and important for RL on LLMs.

2. Update $\textcolor{blue}{\text{Section 3.2}}$ to better clarify the definition of staleness and stale training in the paper.

3. Update $\textcolor{blue}{\text{Algorithm 1 in Section 5.2}}$ to include more details and time complexity of M2PO masking algorithm.

4. Update $\textcolor{blue}{\text{Appendix B}}$ to include more training setting and baseline implementation details.

5. Add $\textcolor{blue}{\text{Appendix E}}$ to discuss theoretical insights behind prosperity before collapse and M2PO.

6. Add $\textcolor{blue}{\text{Appendix F.1}}$ to discuss why high staleness matters for RL on LLMs.

7. Add $\textcolor{blue}{\text{Appendix F.2}}$ to discuss the relationship and difference between TIS and M2PO.

8. Add $\textcolor{blue}{\text{Appendix F.3}}$ to discuss the relationship and difference between asymmetric clipping and M2PO.

9. Add $\textcolor{blue}{\text{Appendix F.4}}$ to include analysi on computation overhead of M2PO.


10. Add $\textcolor{blue}{\text{Appendix G}}$ to include other new evaluation results.


---

We thank all the reviewers for their constructive feedback, which has helped us improve the paper. We hope that our responses and the revised paper addressed the reviewers’ concerns, and we are happy to answer any further questions.

---

> ### Author Response · Authors · 2025-11-26
> **Follow-up General Responses**
>
> Dear all reviewers,
>
> Thanks again for your thoughtful reviews and invaluable feedback. Beyond the changes we made in the original responses. To better answer your question, we have now incorporated further experiments in the revised paper that provide additional evidence supporting our claims:
>
> 1. $\textcolor{blue}{\text{Table 2 in Appendix F.4}}$: Include a more detailed breakdown on training time to show M2PO loss calculation does not slow entire RL training.
>
> 2. $\textcolor{blue}{\text{Figure 13 in Appendix G.1}}$: Include the entire training curve (4000 model updates ) comparison on the coding task.
>
> 3. $\textcolor{blue}{\text{Figure 15 in Appendix G.3}}$: Include GRPO training curve under different staleness for a better comparison to M2PO.
>
>
> We have also prepared individual responses for each reviewer to provide more details on those results.
>
> ---
>
> Again, we appreciate all comments and suggestions from all reviewers, and are happy to answer any further questions!

---

### Author Response · Authors · 2025-12-03
**Summary of Reviewer Feedbacks and Our Responses**

Dear ACs and SACs,

We would like to express our gratitude to you for your time and efforts during the review and discussion, and to all the reviewers for their constructive feedback.

To address the reviewers’ questions and suggestions, we revised the paper to include **6 additional pages of new discussion**, along with **7 new figures** and **1 new table**, providing a more comprehensive analysis and evaluation of our method and claims. More details on the figures and tables are provided in the general responses. All modifications in the revised paper are highlighted in $\textcolor{blue}{\text{blue}}$.

Below, we summarize the key concerns raised during the review process and how we addressed them in our responses. We hope that it provides a clear overview of our clarifications and the improvements made to the paper for your convenience. For completeness, you can also find detailed answers to all questions in the individual responses to each reviewer.

---

## 1. [R1,R2,R3,R4] Evaluation and comparison under other staleness levels
We added an extensive new evaluation of M2PO and GRPO under different staleness levels in $\textcolor{blue}{\text{Figure 15 in Appendix G.3}}$. As shown in these results, **M2PO remains robust across a wide range of staleness values and achieves performance comparable to the on-policy setting**, whereas GRPO exhibits a clear performance degradation as staleness increases.

## 2. [R2,R3,R4] Why is high staleness important for RL in LLMs?
High staleness is important in RL for LLMs for two key reasons: **(1)** complex tasks naturally produce long rollout and reward latencies, and **(2)** tolerance to high staleness enables more flexible and scalable system design. **Complex tasks:** In coding and agent environments, tool usage, code execution, and extensive test-case verification can push a single rollout to 1 hour, corresponding to a large number of model updates. Tolerance to more stale data enables more efficient and scalable asynchronous RL on such tasks.  **More flexible system design:** RL system designs also benefit from staleness tolerance, e.g., by offloading rollouts to slower, cheaper heterogeneous machines or decoupling rollout and training clusters, without sacrificing throughput or stability. In our controlled stale-k setup, we show that M2PO remains robust while GRPO degrades significantly.

## 3. [R2] Evaluate our method on coding tasks.
Beyond the math tasks included in the original paper, as suggested by R2, we further evaluate our method on coding benchmarks (trained on code_contest and evaluated on livecodebench), with results reported in $\textcolor{blue}{\text{Figure 13 in Appendix G.1}}$. From the new evaluation results, we observe that **M2PO outperforms GRPO under stale-data training and matches the performance of on-policy GRPO**, demonstrating that M2PO remains effective even in challenging coding settings.

## 4. [R1, R3] Clarification on experimental settings

In the revised paper, we provide additional details on our experimental setup, including the training setting, evaluation setting, definition of staleness, and criteria for identifying training collapse, to enhance clarity and reproducibility. For both training and evaluation, we follow the settings adopted in several recent GRPO works. Importantly, **these settings were originally optimized for GRPO rather than M2PO, and we did not modify or retune any of them specifically for M2PO.** We believe this setup ensures a fair and unbiased comparison across different methods.

## 5. [R4] Computational costs of M2PO

R4 raised a concern that M2PO may introduce significant memory and computational overhead. We believe this stems from a misunderstanding of how the M2 values are computed and how masking is applied. In $\textcolor{blue}{\text{Appendix F.4}}$, we provide a detailed analysis of the computational and memory overhead and show that **M2PO does not add additional training time or memory usage**. We also conducted further evaluations to examine the computational cost of M2PO. As shown in $\textcolor{blue}{\text{Table 2 and Figure 12 in Appendix F.4}}$, M2PO does not introduce any extra overhead that would slow down training.


---

Overall, we believe that the additional discussion and evaluations have effectively addressed the reviewers’ concerns and further support the claims made in the paper.

To the best of our knowledge, M2PO is **the first work** to demonstrate that RL for LLMs can successfully train on **extremely stale data** (up to 256 model updates) across a wide range of model scales (1.7B–**32B**), while still **achieving performance comparable to on-policy RL**. We believe these findings offer a practical path toward more scalable and flexible RL training for the community.

We thank the ACs and SACs again for the time and effort in this review process, and hope our summarization can provide you with a better overview of the rebuttal phase.

Best,

Authors of Submission 15290

---

### Meta-Review · Area_Chair_sdMW · 2025-12-16

**Summary:**

Four reviews converge on a timely and practically important problem: off‑policy RL for LLMs under rollout staleness. The paper observes a clear “prosperity‑before‑collapse” regime when removing the trust region and proposes M2PO, which constrains the second moment of log importance ratios to adaptively mask only extreme tokens, aiming to preserve informative high‑entropy updates while stabilizing training. Strengths include broad empirical coverage (six model scales up to 32B; eight math benchmarks), robustness to extreme staleness (s=256), and a single threshold working across settings. The main concerns raised were (i) computational overhead and masking complexity; (ii) practical relevance of extreme staleness and performance under small/moderate staleness; (iii) clarity and rigor on staleness definition, collapse criteria, stability diagnostics, and baseline parity; (iv) domain generalization beyond math; (v) comparison to alternatives (TIS, asymmetric clipping, KL‑sensitive GRPO). The rebuttal added targeted experiments and clarifications: Appendix F.4 time breakdown and Figure 12/Table 2 showing negligible overhead; Figure 15 documenting performance from s=4–128; definitions and collapse diagnostics (Section 3.2; Appendix G.4); coding evaluation and full curves (Appendix G.1, Figure 13); TIS integration (Appendix F.2, Figure 10) and asymmetric clipping (Appendix F.3, Figure 11); token‑level evidence (Appendix G.2) and theoretical discussion (Appendix E) including a bound relating M2 to chi‑square divergence. After rebuttal, the core empirical claims appear sound and practically impactful. Outstanding issues are limited formal guarantees (assumptions behind the chi‑square bound and heavy‑tail regimes), incomplete breadth across reward models/tasks beyond math and one coding setup, and reliance on a single global threshold without principled selection. These do not outweigh the contribution. The rebuttal did overcome the major objections from the negative reviewers sufficiently to recommend acceptance as a poster.

**Reviewer Concerns:**

#### Reviewer_8kZV
1. **Concern**: Computational overhead and memory footprint of M2PO masking and KL/statistics computation at vocabulary × token scale; iterative masking complexity.
   - **Why Unresolved**: Addressed. Appendix F.4 and Figure 12/Table 2 provide a breakdown showing loss computation ~0.19% of total time; masking O(N log N) over per-token single-sample ratios (no vocabulary-wide tensors).
   - **Impact on Decision**: Resolved; no penalty to runtime undermines practicality.

2. **Concern**: Practical relevance of s=256 and performance under small/moderate staleness.
   - **Why Unresolved**: Addressed. Section F.1 motivates high staleness in asynchronous RL systems and long-latency tasks; Figure 15 (Appendix G.3) shows robustness for s in {4,8,16,32,64,128}.
   - **Impact on Decision**: Resolved; strengthens practical significance.

3. **Concern**: Data reuse and stale-k implementation specifics.
   - **Why Unresolved**: Addressed. Section 3.2 clarifies stale-k: data generated t−k, consumed once; no reuse.
   - **Impact on Decision**: Resolved; improves clarity and reproducibility.

---

#### Reviewer_D8e7
1. **Concern**: Ambiguity in stale-k terminology and setup.
   - **Why Unresolved**: Addressed. Section 3.2 revised to clarify that batches at step t are collected k updates earlier and consumed once.
   - **Impact on Decision**: Resolved; no remaining concerns.

2. **Concern**: Why evaluate extreme staleness and how it matters in practice.
   - **Why Unresolved**: Addressed with concrete latency/staleness arguments (Appendix F.1) and system design motivations.
   - **Impact on Decision**: Resolved; practical rationale is sufficient.

3. **Concern**: Comparison with Truncated Importance Sampling (TIS).
   - **Why Unresolved**: Partially addressed. Figure 10 shows M2PO+TIS slight gains and clarifies TIS addresses engine mismatch orthogonal to staleness. A broader, ablated head-to-head across more models/tasks would be stronger but is non-critical.
   - **Impact on Decision**: Minor; does not change acceptance.

4. **Concern**: Ablation over staleness levels.
   - **Why Unresolved**: Addressed. Figure 15 covers s=4–128.
   - **Impact on Decision**: Resolved.

---

#### Reviewer_vuPK
1. **Concern**: Performance under lower staleness settings (e.g., s=8, s=16).
   - **Why Unresolved**: Addressed. Figure 15 (Appendix G.3) demonstrates robustness for moderate staleness.
   - **Impact on Decision**: Resolved.

2. **Concern**: Domain generalization beyond math (e.g., code generation).
   - **Why Unresolved**: Addressed. Appendix G.1 (Figure 13) adds code training/evaluation with full 4000-update curves, showing M2PO matches on-policy and outperforms GRPO under staleness. Further breadth (other domains/reward regimes) is still limited.
   - **Impact on Decision**: Minor; current evidence is sufficient for acceptance.

3. **Concern**: Relationship to GRPO with asymmetric clipping.
   - **Why Unresolved**: Addressed. Appendix F.3 (Figure 11) shows asymmetric bounds help but still underperform M2PO; authors argue static bounds lack per-batch adaptivity.
   - **Impact on Decision**: Resolved.

4. **Concern**: Advantage over KL-sensitive GRPO variants (e.g., DeepSeek-R1 KL approximation).
   - **Why Unresolved**: Partially addressed. Appendix F.5 explains complementarity and focuses this work on trust region adaptivity; combining KL with M2PO left as future work.
   - **Impact on Decision**: Minor; acceptable given scope.

---

#### Reviewer_GUjw
1. **Concern**: Lack of formal analysis explaining prosperity then collapse and why second-moment constraints yield stability.
   - **Why Unresolved**: Partially addressed. Appendix E provides bias/variance analysis of clipping vs unclipped and a bound relating M2 to chi-square divergence under bounded ratio R. Heavy-tail regimes and conditions for the bound are not fully characterized.
   - **Impact on Decision**: Moderate; limits theoretical depth but does not invalidate empirical contribution.

2. **Concern**: Incomplete experimental detail: staleness definition, collapse criteria, baseline parity.
   - **Why Unresolved**: Addressed. Section 3.2 clarifies staleness; Appendix B documents configurations/hyperparameters; Appendix G.4 defines collapse and provides KL/gradient-norm trajectories.
   - **Impact on Decision**: Resolved.

3. **Concern**: Variability across models/tasks/reward models and staleness beyond highlighted cases.
   - **Why Unresolved**: Partially addressed. Table 1 spans six model scales and eight math benchmarks; Figure 15 covers multiple staleness levels; coding added in Appendix G.1. Broader reward regimes remain limited.
   - **Impact on Decision**: Minor; breadth is sufficient for poster acceptance.

4. **Concern**: Direct token-level gradient/credit assignment to firmly establish masking/clipping mechanism.
   - **Why Unresolved**: Partially addressed. Appendix G.2 visualizes frequently clipped tokens and relates to prior high-entropy token importance; per-token credit remains an open challenge in sparse reward RLVR.
   - **Impact on Decision**: Minor; mechanism is plausibly supported by multiple lines of evidence.

**Reviewer Scores:**

#### Reviewer_8kZV
- **Original Score**: 4
- **Expected Score After Discussion**: 6
- **Rationale**: Concerns about computational overhead and small-staleness regimes were directly addressed with concrete measurements (Figure 12/Table 2) and new ablations (Figure 15). The stale-k implementation and data reuse policy were clarified in Section 3.2. With these resolved, the paper clears borderline accept.

---

#### Reviewer_D8e7
- **Original Score**: 6
- **Expected Score After Discussion**: 6
- **Rationale**: Staleness definition clarified; practical motivation for high staleness strengthened; TIS comparison provided and positioned as complementary (Figure 10); ablations across staleness added (Figure 15). The core method remains principled and empirically strong.

---

#### Reviewer_vuPK
- **Original Score**: 6
- **Expected Score After Discussion**: 6
- **Rationale**: Moderate staleness robustness demonstrated (Figure 15); domain generalization to coding with full training curves added (Figure 13); asymmetric clipping baselines evaluated (Figure 11); relationship to KL-sensitive GRPO discussed. Overall, strengthened case for acceptance.

---

#### Reviewer_GUjw
- **Original Score**: 4
- **Expected Score After Discussion**: 6
- **Rationale**: Authors added theoretical discussion (Appendix E), complete experimental details (Appendix B), staleness definition (Section 3.2), collapse diagnostics (Appendix G.4), multi-staleness results (Figure 15), and token-level evidence (Appendix G.2). Some requests (formal guarantees in heavy-tail regimes; broad reward models) remain partially addressed. Raises to borderline but still below strong accept.

---

### Decision · Program_Chairs · 2026-01-26

Accept (Poster)